# FASTER CASCADES VIA SPECULATIVE DECODING

**Harikrishna Narasimhan**[1], **Wittawat Jitkrittum**[1], **Ankit Singh Rawat**[1]
**Seungyeon Kim**[2], **Neha Gupta**[3,†], **Aditya Krishna Menon**[1], **Sanjiv Kumar**[1]
[1]Google Research,  [2]Google DeepMind,  [3]Mistral AI
{hnarasimhan, wittawat, ankitsrawat, adityakmenon, sanjivk}@google.com

## ABSTRACT

Cascades and speculative decoding are two common approaches to improving language models' inference efficiency. Both approaches interleave two models of different sizes, but via fundamentally distinct mechanisms: cascades employ a *deferral rule* that invokes the larger model only for "hard" inputs, while speculative decoding uses *speculative execution* to primarily invoke the larger model in parallel scoring mode. These mechanisms offer different benefits: cascades offer compelling cost-quality trade-offs, often even outperforming the large model; speculative cascades offer impressive speed-ups, while guaranteeing quality-neutrality. In this paper, we leverage the best of both these approaches by designing new *speculative cascading* techniques that implement their deferral rule through speculative execution. We characterize the optimal deferral rule for our speculative cascades, and employ a plug-in approximation to the optimal rule. Experiments with Gemma and T5 models on a range of language benchmarks show that our approach yields better cost-quality trade-offs than cascading and speculative decoding baselines.

## 1 INTRODUCTION

Large language models (LLMs) have yielded significant advances in quality on a range of natural language processing tasks (Radford et al., 2018; Raffel et al., 2020; Brown et al., 2020; Black et al., 2022; Chowdhery et al., 2022; Anil & et al., 2023; Touvron et al., 2023; Team et al., 2023; et al., 2024b;a), at the cost of an increase in inference latency. This has sparked a growing body of literature on reducing LLMs' inference costs without (overly) compromising on quality (Elbayad et al., 2020; Pope et al., 2022; Schuster et al., 2022; Leviathan et al., 2023; Chen et al., 2023a; Sheng et al., 2023; Sun et al., 2024). One such line of work involves constructing a family of models of various sizes (e.g., a small and large model), and suitably orchestrating amongst them to make a prediction. Two canonical instantiations of this strategy are *model cascading* (Wang et al., 2020; Mamou et al., 2022; Varshney & Baral, 2022; Khalili et al., 2022; Dohan et al., 2022; Chen et al., 2023b; Gupta et al., 2024; Ding et al., 2024) and *speculative decoding* (Stern et al., 2018; Chen et al., 2023a; Leviathan et al., 2023; Sun et al., 2024; Li et al., 2024a; Xia et al., 2024).

While similar in spirit, cascades and speculative decoding are fundamentally different in details. Cascades employ a *deferral rule* to identify "hard" inputs, and only invoke larger models on such inputs. For example, in a two-model cascade, one first invokes the smaller model, and uses its associated probability of the generated output to decide whether to defer to the larger model. By contrast, speculative decoding uses a small model to *draft* a block of tokens via standard autoregressive decoding, which are then *verified* in parallel by a large model. One then accepts all drafted tokens until the first "implausible" one, which is rolled back based on the larger LM's prediction.

Owing to their different mechanisms, both methods have complementary strengths. Cascades seek to output distributions that have the best quality for a given cost budget, sometimes even yielding better quality than the individual models they are constructed with (Jitkrittum et al., 2023; Kim et al., 2023) (§3). By contrast, speculative decoding is theoretically guaranteed to match the output distribution (or a close approximation thereof (Tran-Thien, 2023)), and is practically observed to provide impressive speed-ups (Stern et al., 2018; Chen et al., 2023a; Leviathan et al., 2023; Sun et al., 2024). Given their complementary nature, a natural question arises: *can we leverage the best of both techniques?*

---

†Work done while working at Google.

In this paper, we do so by designing new techniques for two-model cascades that implement their deferral rule in a speculative manner: we have the smaller model generate drafts auto-regressively, and the larger model execute in parallel on the drafts to decide *whether or not to defer on them.* We show that this *speculative cascading* approach yields better cost-quality trade-offs

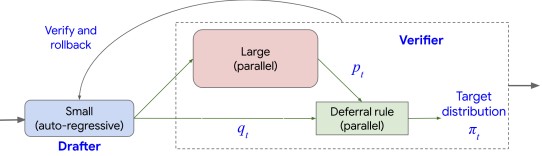

Figure 1: Speculative cascade inference between a small and a large LM via a *deferral rule*.

than both standard cascades and speculative decoding. In detail, we make the following contributions:

(i) We introduce a general recipe for speculative execution, where we seek to mimic a general *target* distribution that interleaves the drafter's and verifier's distributions. Lossy speculative sampling (Tran-Thien, 2023) is a special case of this recipe for a particular target distribution (§4.1).

(ii) We show how common cascading rules, such as Chow's rule (Chow, 1970) and confidence-difference thresholding (Jitkrittum et al., 2023), can be implemented speculatively by plugging in their target distribution into our framework. We refer to these as *speculative cascades* (§4.2).

(iii) We characterize the *theoretically optimal* deferral rule for a speculative cascade, and design a speculative cascading technique that implements a plug-in estimate to the optimal rule (§4.3, Lemma 4, Table 1). We also present token-specific variants of our deferral rules (§5).

(iv) Through experiments with Gemma (Team et al., 2024) and T5 models (Raffel et al., 2020) on a range of benchmark language tasks including summarization, translation, reasoning, coding and QA, we show that speculative cascades are able to provide better cost-quality trade-offs than their sequential cascade and speculative decoding counterparts (§6).

Overall, we aim to develop a principled approach to trade-off quality and inference costs by interleaving two models of different sizes, with promising empirical results. We hope to inspire future research adapting the proposed ideas with ingredients underpinning the state-of-the-art in speculative decoding (Cai et al., 2024; Li et al., 2024a;b; Chen et al., 2024).

## 2    A TALE OF TWO EFFICIENT LM INFERENCE STRATEGIES

Let $\mathcal{V}$ denote a finite vocabulary of *tokens*, with $\mathcal{V}^*$ denoting the set of all finite-length *sequences* generated by this vocabulary. Let $\Delta_{\mathcal{V}}$ denote the set of all probability distributions over tokens in $\mathcal{V}$. Given an arbitrary length sequence $x = x_1 x_2 \ldots x_L \in \mathcal{V}^*$ and index $i \leq L$, we denote $x_{<i} = x_1 x_2 \ldots x_{i-1}$. A *language model* (LM) is a probability distribution over $\mathcal{V}^*$. Let $\mathbb{P}$ denote the data generating probability distribution over $\mathcal{V}^*$. This could be, for example, a distribution over prompt-response pairs that the LM may encounter during deployment, or a distribution of sequences used to pre-train the LM. We will measure the quality of an LM based on how closely it mimics $\mathbb{P}$.

Suppose we are provided two LMs $q$ and $p$, where $p$ is the larger (more expensive) model. Our goal is to design an inference strategy that selectively invokes $q$ and $p$ to trade-off between quality and latency (which may be approximated by the fraction of times that $p$ is invoked). We will denote by $q(x_t | x_{<t})$ the probability $q$ associates to token $x_t \in \mathcal{V}$ given prefix $x_{<t} \in \mathcal{V}^{t-1}$, and by $p(x_t | x_{<t})$ the same distribution from model $p$. Whenever it is clear from context, we will hide the conditioning on prefix $x_{<t}$, and use the shorthand $q_t(\cdot)$ for $q(\cdot | x_{<t})$ and $p_t(\cdot)$ for $p(\cdot | x_{<t})$.

**Cascades** are an effective strategy to trade-off cost and quality by having the smaller model $q$ handle the "easy" samples, and the larger model $p$ handle the "hard" ones (Gupta et al., 2024; Yue et al., 2024). A common cascading approach is confidence thresholding or Chow's rule (Chow, 1970; Jitkrittum et al., 2023), where we first run $q$ on the input, and defer to $p$ when $q$'s confidence for its generated response is sufficiently low. This strategy is typically implemented at the *sequence-level*, where for a given prefix $x_{<m}$ we invoke $q$, evaluate its maximum conditional probability over all responses, and check whether it falls below a threshold $\alpha \in [0, 1]$:

$$\max_{x_m \ldots x_{m+n}} q(x_m \ldots x_{m+n} \,|\, x_{<m}) < 1 - \alpha. \tag{1}$$

If this holds, we defer to $p$ to generate a new response; otherwise, we generate a response with $q$. One may tune $\alpha$ to achieve a desired cost-quality trade-off. The literature also offers variants of Chow's rule that use a more nuanced aggregation of *per-token* uncertainties (Gupta et al., 2024).

Table 1: Target distributions associated with different inference algorithms, where $\alpha$ is a free parameter and $\beta \geq 1 - \alpha$ depends on $\alpha$, $q$ and $p$. The last column indicates whether the execution is sequential (Algorithm 2), via an oracle (Algorithm 3), or speculative (Algorithm 5) (Leviathan et al., 2023). See (6) for details on $\delta$. The third row presents a variant of the BiLD algorithm of Kim et al. (2023), where $D(q, p)$ is a measure of discrepancy between $q$ and $p$; the original algorithm differs in the use of a deterministic speculative decoding procedure with a dynamic draft window (see §B).

| Inference strategy | Deferral decision $\delta(q, p)$ | Target distribution $\pi(u)$ | Execution |
|---|---|---|---|
| SpecDecoding (Leviathan et al., 2023) | - | $p(u)$ | Speculative |
| Lossy SpecDecoding (Tran-Thien, 2023) | - | $\max\{\min\{q(u), \frac{p(u)}{1-\alpha}\}, \frac{p(u)}{\beta}\}$ | Speculative |
| BiLD* variant (Kim et al., 2023) | $\mathbf{1}\big(D(q,p) > \alpha\big)$ | $(1-\delta) \cdot q(u) + \delta \cdot p(u)$ | Speculative |
| TokenCascade [Chow] (Chow, 1970) | $\mathbf{1}\big(\max_v q(v) < 1 - \alpha\big)$ | $(1-\delta) \cdot q(u) + \delta \cdot p(u)$ | Sequential |
| Oracle [Diff] (Jitkrittum et al., 2023) | $\mathbf{1}\big(\max_v q(v) < \max_v p(v) - \alpha\big)$ | $(1-\delta) \cdot q(u) + \delta \cdot p(u)$ | Oracle |
| SpecCascade [Chow] | $\mathbf{1}\big(\max_v q(v) < 1 - \alpha\big)$ | $(1-\delta) \cdot q(u) + \delta \cdot p(u)$ | Speculative |
| SpecCascade [Diff] | $\mathbf{1}\big(\max_v q(v) < \max_v p(v) - \alpha\big)$ | $(1-\delta) \cdot q(u) + \delta \cdot p(u)$ | Speculative |
| SpecCascade [OPT] | $\mathbf{1}\big(\max_v q(v) < \max_v p(v) - \alpha \cdot D_{\mathrm{TV}}(p,q)\big)$ | $(1-\delta) \cdot q(u) + \delta \cdot p(u)$ | Speculative |

**Speculative decoding** is an alternate strategy that applies *token-level* interleaving between $q$ and $p$, seeking to *provably* match the larger model quality at a reduced inference cost (Stern et al., 2018; Leviathan et al., 2023). Given a prefix $x_{<t}$, we *draft* $\gamma$ draft tokens $x_t, \ldots, x_{t+\gamma-1}$ via auto-regressive sampling from $q$, and *verify* if these tokens can be accepted by running $p$ in parallel on the $\gamma$ prefixes $x_{<t}, \ldots, x_{<t+\gamma-1}$. We then rollback to the first rejected token $t + j^*$ (where $j^* \in \{0, 1, \ldots, \gamma - 1\}$), replace $x_{t+j^*}$ with a new token, and repeat the process with prefix $x_{<t+j^*+1}$.

During the verification stage, a draft token $x_{t+j}$ generated by $q$ is accepted with probability $\min\left(1, \frac{p_{t+j}(x_{t+j})}{q_{t+j}(x_{t+j})}\right)$ and rejected otherwise, recalling the shorthand $q_{t+j}(\cdot) = q(\cdot|x_{<t+j})$ and $p_{t+j}(\cdot) = p(\cdot|x_{<t+j})$. A rejected token is then replaced by a new token sampled from a modified distribution $\mathrm{norm}\left(\max\{0, p_{t+j}(\cdot) - q_{t+j}(\cdot)\}\right)$, where $\mathrm{norm}(\cdot)$ denotes normalization to sum to 1. This sampling process is provably equivalent to sampling $\gamma$ tokens auto-regressively from $p$ for prefix $x_{<t}$ (Leviathan et al., 2023). We summarize this speculative sampling procedure in Algorithm 1. Each invocation of this algorithm generates at most $\gamma + 1$ next tokens (and at least one) for a given prefix $x_{<t}$. One may run this algorithm multiple times to generate a complete output sequence.

In practice, one may employ a lossy variant (Tran-Thien, 2023) of the above sampling that allows some *deviation* from verifier's distribution $p$. In this case, a draft token $x_{t+j}$ is accepted with probability $\min\left(1, \frac{p_{t+j}(x_{t+j})}{(1-\alpha) \cdot q_{t+j}(x_{t+j})}\right)$, where $\alpha \in [0, 1)$ is a strictness parameter, with higher values indicating greater deviation from $p$. A rejected token may then be replaced by a token sampled from the residual distribution $\mathrm{norm}\left(\max\left\{0, \frac{1}{\beta} \cdot p_{t+j}(\cdot) - q_{t+j}(\cdot)\right\}\right)$, where $\beta \geq 1 - \alpha$ is a parameter that depends on $\alpha$, $q$ and $p$. A common heuristic is to simply set $\beta = 1$ (Zhou et al., 2024).

## 3 CASCADES MEET SPECULATIVE DECODING

Both cascades and speculative decoding interleave models of different sizes to reduce inference cost, but fundamentally differ in the mechanisms they use. As a step towards comparing the strengths and weaknesses of these approaches, we first describe how one may design a *token-level cascade*.

### 3.1 WARM-UP: TOKEN-LEVEL CASCADES

It is straightforward to extend the sequence-level Chow's rule from §2 to form a *token-level cascade* between $q$ and $p$. For a prefix $x_{<t}$, we first compute the smaller model's distribution $q_t(\cdot)$, and check whether $\max_{v \in \mathcal{V}} q_t(v)$ is below a pre-chosen threshold. If so, we evaluate $p_t(\cdot)$, and sample $x_t \sim p_t(\cdot)$; otherwise, we sample $x_t \sim q_t(\cdot)$.

More generally, we may design a token-level *deferral rule* $r : \mathcal{V}^{t-1} \to \{0, 1\}$ that takes the prefix $x_{<t}$ as input and outputs a binary decision, with $r(x_{<t}) = 1$ indicating that we defer to $p$ (i.e., draw a sample from $p$ rather than $q$). For example, token-level Chow's rule can be written as:

$$r_{\mathtt{Chow}}(x_{<t}) = 1 \iff \max_{v \in \mathcal{V}} q_t(v) < 1 - \alpha, \tag{2}$$

where $\alpha$ is a threshold parameter; the higher the value, the lower is the frequency of deferral to $p$. One may also use other confidence measures than the maximum probability, such as the entropy of the small model's probability distribution. We elaborate in §D.1 that the choice of confidence measure would depend on the evaluation metric of interest; Equation (2) is typically prescribed when the cascade's quality is evaluated in terms of its accuracy against the data generating distribution on individual tokens, whereas entropy is prescribed when the metric of interest is the cross-entropy loss.

## 3.2 OPTIMAL TOKEN-LEVEL CASCADE DEFERRAL

While Chow's rule (2) is easy to implement, it can be sub-optimal if the smaller model's max-token probability is not reflective of which of the two models are better equipped to predict the next token for a given prefix (Jitkrittum et al., 2023). Given this, it is natural to ask what the *optimal* deferral rule $r$ for a token-cascade looks like, and whether we can reasonably approximate this rule.

For this, we must first specify an objective to minimize at each step $t$. Following the prior cascade literature (Jitkrittum et al., 2023; Gupta et al., 2024), a reasonable objective to minimize is the expected loss from the deferral rule against the data generating distribution $\mathbb{P}$, with an added cost for deferring to the larger model. We state this below for a fixed prefix $x_{<t}$, using as before the short-hand $q_t(\cdot)$ for $q(\cdot|x_{<t})$ and $p_t(\cdot)$ for $p(\cdot|x_{<t})$:

$$L_{\text{def}}(r; x_{<t}) = \mathbb{E}_{v \sim \mathbb{P}(\cdot|x_{<t})}\Big[\big(1 - r(x_{<t})\big) \cdot \ell(v, q_t) + r(x_{<t}) \cdot \big(\ell(v, p_t) + \alpha\big)\Big], \quad (3)$$

for a cost penalty $\alpha \geq 0$ and loss function $\ell : \mathcal{V} \times \Delta_{\mathcal{V}} \rightarrow \mathbb{R}_+$. Common choices for $\ell$ include the 0-1 loss $\ell_{\text{0-1}}(v, q_t) = \mathbf{1}\left(v \neq \arg\max_{v'} q_t(v')\right)$ and the log loss $\ell_{\log}(v, q_t) = -\log\left(q_t(v)\right)$.

**Lemma 1** (Optimal deferral for token-level cascades (Jitkrittum et al., 2023)). *The minimizer of (3) is of the form:*

$$r^*(x_{<t}) = 1 \iff \mathbb{E}_{v \sim \mathbb{P}(\cdot|x_{<t})}\left[\ell(v, q_t)\right] > \mathbb{E}_{v \sim \mathbb{P}(\cdot|x_{<t})}\left[\ell(v, p_t)\right] + \alpha. \quad (4)$$

Intuitively, we compare the expected loss from $q$ with the expected cost of invoking $p$, and decide to defer when the latter is smaller. We note here that this optimization problem is set up for a *fixed* prefix $x_{<t}$. One may also consider the coupled optimization problem across all positions.

*Plug-in estimator for (4).* The optimal rule in (4) requires computing expectations over the data generating distribution $\mathbb{P}(\cdot|x_{>t})$, which is not available during inference time. A common approach in the cascades literature is to replace the expected losses with the models' confidence estimates (Jitkrittum et al., 2023). For example, when $\ell = \ell_{\text{0-1}}$, it may be reasonable to use $1 - \max_v q_t(v)$ as an estimate of the expected 0-1 loss $\mathbb{E}_{x_t \sim \mathbb{P}(\cdot|x_{<t})}\left[\ell_{\text{0-1}}(x_t, q_t)\right]$ and $1 - \max_v p_t(v)$ as an estimate of $\mathbb{E}_{x_t \sim \mathbb{P}(\cdot|x_{<t})}\left[\ell_{\text{0-1}}(x_t, p_t)\right]$. The extent to which these estimates are accurate depend on how well $q$ and $p$ are calibrated (Guo et al., 2017). The resulting plug-in estimator for (4) thresholds the *difference* of confidence estimates from both distributions:

$$\boxed{\hat{r}_{\texttt{Diff}}(x_{<t}) = 1 \iff \max_v q_t(v) < \max_v p_t(v) - \alpha.} \quad (5)$$

Similarly, when $\ell = \ell_{\log}$, we may use the entropy $-\sum_v q_t(v) \cdot \log(q_t(v))$ from $q_t$ as an estimate of its expected log-loss, and similarly for $p_t$ (see § D).

**Remark 1** (`Diff` rule is not realizable with a token-level cascade). *We cannot directly employ $\hat{r}_{\texttt{Diff}}$ in a token-level cascade, as it needs the large model to be invoked at every step $t$. However, it serves as an* oracle *that allows to analyze the head-room available to improve upon Chow's rule.*

## 3.3 CONTRASTING TOKEN-LEVEL CASCADE AND SPECULATIVE DECODING TRADE-OFFS

Token-level cascades and speculative decoding differ in the distribution over tokens they seek to mimic. Speculative decoding seeks to mimic the large model's output distribution, and is usually used when one wants to match the quality of the large model. On the other hand, token-level cascades seek to output distributions that closely approximate the label distribution and potentially offer *good cost-quality trade-offs*, sometimes yielding better quality than even the large model.

Cascades are useful when the draft model fares better than the verifier on some inputs, and one may want to retain the drafter's predictions even when it disagrees with the verifier. Even in cases where

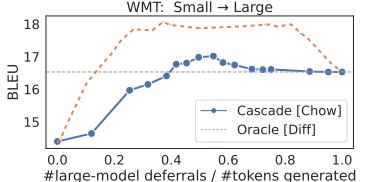 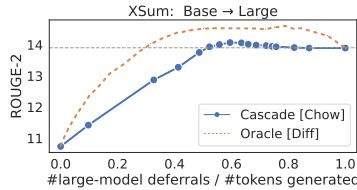

Figure 2: Plots of quality as a function of the *number of deferrals to the larger model divided by the total number of generated tokens* for cascades constructed from T5 models (under temperature sampling with $T = 1$). The left-most point represents the small model and the right-most represents the large model. We compare token-level cascades constructed with Chow's rule (`Chow`) and an oracle deferral rule (`Diff`). While speculative decoding will match the quality of the large model (see dashed horizontal line), the oracle deferral rule yields significantly better quality on a range of deferral rates.

both the drafter and verifier fare poorly on some inputs (e.g., due to label noise), one may want to ignore the disagreement between the drafter and verifier to avoid triggering unnecessary roll-backs.

As a concrete example, we consider token-level cascades of T5 models (Raffel et al., 2020) of two different sizes finetuned on a WMT EN → DE translation Bojar et al. (2014) and an extreme summarization (XSum) task (Narayan et al., 2018). We construct these cascades using Chow's rule in (2), and the `Diff` rule in (5), which as noted in Remark 1, serves as an oracle. In Figure 2, we plot quality as a function of fraction of samples deferred to the large model (number of deferrals divided by number of generated tokens), as we vary the cost parameter $\alpha$. Recall that speculative decoding is guaranteed to match the quality of the large model. In contrast, the `Diff` rule based cascades yield a wide range of cost-quality trade-offs, often outperforming the large model. Even Chow's rule, which is sub-optimal for cascading (Jitkrittum et al., 2023), outperforms the large model in a small region. As noted by Kim et al. (2023), this may be attributed to the ensembling effect in a cascade.

However, compared to speculative decoding, token-level cascades may require a significantly larger number of deferrals to the large model to achieve the same quality. This is because token-level cascades are executed *sequentially*: whenever $q$ defers, we execute $p$ once to generate one next token for the prefix accumulated so far, and the control transfers back to $q$. In contrast, speculative decoding runs $p$ in *scoring* mode to verify $\gamma$ draft tokens from $q$ in *parallel*. Moreover, the stochastic verification algorithm in speculative decoding often results in fewer tokens from $q$ getting rejected compared to the deterministic deferral rules used in a cascade. These observations motivate a natural question: *given their complementary strengths, how can we leverage the best of both these techniques?*

## 4 SPECULATIVE CASCADES: LEVERAGING THE BEST OF BOTH WORLDS

In addressing the above question, we present our main contribution: *speculative cascades*, a principled approach to combining the better trade-offs token-level cascades offer with the faster execution of speculative decoding. Unlike token-level cascades, where the large model is called only when the small model defers, speculative cascades invoke the large model in parallel scoring mode after every $\gamma$ draft tokens. Consequently, they have the added benefit of being able to implement deferral rules that are not realizable with a sequential cascade, and can thus potentially achieve lower latencies.

### 4.1 SPECULATIVE DECODING WITH GENERAL TARGET DISTRIBUTIONS

We begin by considering a generic version of speculative decoding that seeks to mimic a *general* target distribution derived from the drafter's and verifier's distributions. In the proposed sampling procedure outlined in Algorithm 4, we sample tokens auto-regressively as before from the drafter's distribution. During the verification step, however, we do not compare the drafter's token probabilities against the verifier's distribution. Instead, we use a user-specified target distribution $\pi = \mathbb{T}(q, p) \in \Delta_V$ derived from the drafter's and verifier's distributions at position $t$, for some function $\mathbb{T}(\cdot, \cdot)$ that is inexpensive to compute. We accept a draft token $x_t$ when $q(x_t) \leq \pi(x_t)$ and reject it otherwise with probability $1 - \frac{\pi(x_t)}{q(x_t)}$. Upon rejection, we re-sample from the residual distribution $\text{norm}\left(\max\{0, \pi(\cdot) - q(\cdot)\}\right)$.

This general procedure not only encompasses standard speculative decoding (Leviathan et al., 2023) for $\mathbb{T}(q, p) = p$, but also includes lossy speculative decoding (Tran-Thien, 2023) as a special case:

**Lemma 2.** *Algorithm 4 reduces to the lossy speculative sampling procedure in (Tran-Thien, 2023) with parameters $\alpha$ and $\beta$ when $\mathbb{T}(q, p)(v) = \max\{\min\{q(v), \frac{p(v)}{1-\alpha}\}, \frac{p(v)}{\beta}\}$.*

## 4.2 FROM SEQUENTIAL TO SPECULATIVE CASCADES

Equipped with Algorithm 4, we now propose new cascading techniques that implement their deferral rule in a speculative manner. Recall from §3.1 that a token-level cascade of two models $q$ and $p$ is defined by a deferral rule $r : \mathcal{V}^{t-1} \to \{0, 1\}$. For a prefix $x_{<t}$, the next-token distribution at position $t$ modeled by this cascade can be written as:

$$\pi(v) = (1 - r(x_{<t})) \cdot q_t(v) + r(x_{<t}) \cdot p_t(v).$$

In fact, for all the deferral rules described in §2, the resulting distribution can be described by a target distribution function $\mathbb{T}_\delta$ of the form:

$$\mathbb{T}_\delta(q, p)(v) = (1 - \delta(q, p)) \cdot q(v) + \delta(q, p) \cdot p(v), \qquad (6)$$

for some function $\delta : \Delta_\mathcal{V} \times \Delta_\mathcal{V} \to \{0, 1\}$ that maps distributions $(q, p)$ to a binary decision. For example, to implement Chow's rule, we may choose $\delta(q, p) = \mathbf{1}\left(\max_v q(v) < 1 - \alpha\right)$.

Our proposal is to then *invoke the speculative sampling procedure in Algorithm 4 with $\mathbb{T}_\delta$ as the target distribution function.* We outline this generic *speculative cascading* approach in Algorithm 5, and contrast it with the sequential execution of a deferral rule in Algorithm 2.

Interestingly, as noted below, the `Diff` rule, which was not realizable with a token-level cascade, can be efficiently implemented with a speculative cascaded using $\delta(q, p) = \mathbf{1}\left(\max_v q(v) < \max_v p(v) - \alpha\right)$. See Table 1 for a summary of different deferral rules and corresponding target distributions.

**Remark 2** (`Diff` **rule is realizable with a speculative cascade**). *In a token-level cascade, the large model's distribution $p$ cannot be used at the time the deferral decision is made (see Remark 1), as this would defeat the purpose of the cascade. With a speculative cascade, however, we can employ rules such as `Diff` that depend on both $q$ and $p$. This is because we run the large model $p$ in parallel on drafts generated by the small model $q$, allowing us to compute both $p(\cdot)$ and $q(\cdot)$ on every prefix.*

So far we have considered deferral rules designed for use with (sequential) token-level cascades. In what follows, we derive the optimal deferral rule $r$ for a speculative cascade, where we sample speculatively from a target distribution $\pi = (1 - r(x_{<t})) \cdot q_t + r(x_{<t}) \cdot p_t$ using $q_t$ as the drafter.

## 4.3 OPTIMAL SPECULATIVE CASCADE DEFERRAL

We seek a deferral rule $r : \mathcal{V}^{t-1} \to \{0, 1\}$ that trades-off between quality and inference cost. As with §2, we measure quality in terms of the loss incurred against the data generating distribution. The inference cost, on the other hand, crucially depends on how frequently a draft token is rejected in the verification phase, triggering a rollback. To this end, we seek to minimize the expected loss from the deferral rule subject to a constraint on the resulting rejection rate. More specifically, (i) we show that the rejection rate can be computed using a simple closed-form expression (Lemma 3); (ii) we formulate a constrained optimization objective (7) and the corresponding the Lagrangian (8); (iii) we derive the optimal deferral rule that minimizes the Lagrangian (Lemma 4), approximate it with a plug-in rule (10), and provide a regret bound guarantee for the approximation (Lemma 5).

**Lemma 3.** *For a given prefix $x_{<t}$, and target distribution $\pi = (1 - r(x_{<t})) \cdot q_t + r(x_{<t}) \cdot p_t$, the probability of a token drawn from draft distribution $q_t$ being rejected is equal to: $r(x_{<t}) \cdot D_{\mathrm{TV}}(p_t, q_t)$, where $D_{\mathrm{TV}}(p, q) = \sum_{v \in \mathcal{V}} \max\{0, p(v) - q(v)\}$ is the TV distance between $p$ and $q$.*

Intuitively, whenever $r(x_{<t}) = 0$, $\pi(v) = q_t(v)$, and therefore there is no rejection or roll-back; when $r(x_{<t}) = 1$, the rejection rate equals $D_{\mathrm{TV}}(p_t, q_t)$, per Leviathan et al. (2023).

For a fixed prefix $x_{<t}$, we formulate the goal of finding a solution to:

$$\min_r \ \mathbb{E}_{v \sim \mathbb{P}(\cdot|x_{<t})}\left[\left(1 - r(x_{<t})\right) \cdot \ell(v, q_t) + r(x_{<t}) \cdot \ell(v, p_t)\right] \ \text{s.t.} \ r(x_{<t}) \cdot D_{\mathrm{TV}}(p_t, q_t) \leq B, \quad (7)$$

for some budget $B > 0$. Equivalently, one may minimize an unconstrained objective similar to (3), for suitable cost parameter $\alpha > 0$ (see §D.5):

$$L_{\mathrm{spec}}(r; x_{<t}) = \mathbb{E}_{v \sim \mathbb{P}(\cdot|x_{<t})}\left[\left(1 - r(x_{<t})\right) \cdot \ell(v, q_t) + r(x_{<t}) \cdot \left(\ell(v, p_t) + \alpha \cdot D_{\mathrm{TV}}(p_t, q_t)\right)\right], \quad (8)$$

---

**Algorithm 1** `SpecDecode`

---

**Input:** Models $q, p$, Prefix $x_{<t}$, Block size $\gamma$
  $\mathbb{T}(q, p) \doteq p$
**Output:** `GenSpecSample`$(q, p, \mathbb{T}, x_{<t}, \gamma)$

---

**Algorithm 2** `TokenCascade`

---

**Input:** Models $q, p$, Deferral logic $\delta$, Prefix $x_{<t}$
  $q_t(\cdot) \doteq q(\cdot|x_{<t})$
  **if** $\delta(q_t, \emptyset) = 0$ **then**
    Sample $x_t \sim q_t(\cdot)$
  **else**
    $p_t(\cdot) \doteq p(\cdot|x_{<t});$  Sample $x_t \sim p_t(\cdot)$
  **end if**
**Output:** $x_t$

---

**Algorithm 3** `OracleCascade`

---

**Input:** Models $q, p$, Deferral logic $\delta$, Prefix $x_{<t}$
  $q_t(\cdot) \doteq q(\cdot|x_{<t});$   $p_t(\cdot) \doteq p(\cdot|x_{<t})$
  **if** $\delta(q_t, p_t) = 0$ **then**
    Sample $x_t \sim q_t(\cdot)$
  **else**
    Sample $x_t \sim p_t(\cdot)$
  **end if**
**Output:** $x_t$

---

**Algorithm 4** `GenSpecSample`

---

**Input:** Models $q, p$, Target distr. $\mathbb{T}$, Prefix $x_{<t}$, Block size $\gamma$
  `// Sample γ tokens auto-regressively from q`
  **for** $j = 0$ to $\gamma - 1$ **do**
    $q_{t+j}(\cdot) \doteq q(\cdot|x_{<t+j});$     $x_{t+j} \sim q_{t+j}(\cdot)$
  **end for**
  `// Run p in parallel to score γ draft tokens`
  $p_{t+j}(\cdot) \doteq p(\cdot|x_{<t+j}), \; \forall j \in [\gamma] \equiv \{0, \dots, \gamma\}$
  $\pi_{t+j} = \mathbb{T}(q_{t+j}, p_{t+j})$
  `// Find the earliest rejected draft token`
  $a_j \sim \text{Ber}\left(\min\left\{1, \frac{\pi_{t+j}(x_{t+j})}{q_{t+j}(x_{t+j})}\right\}\right), \; \forall j \in [\gamma - 1]; \quad a_\gamma = 0$
  $j^* = \min\{j \in [\gamma] : a_j = 0\}$
  `// Sample a new token from residual distribution`
  $p_{\text{res}}(\cdot)$
    $= \begin{cases} \text{norm}(\max\{0, \pi_{t+j^*}(\cdot) - q_{t+j^*}(\cdot)\}) & \text{if } j^* < \gamma \\ \pi_{t+\gamma}(\cdot) & \text{else} \end{cases}$
  Sample $x_{t+j^*} \sim p_{\text{res}}(\cdot)$
**Output:** $x_t, \dots, x_{t+j^*}$

---

**Algorithm 5** `SpecCascade`

---

**Input:** Models $q, p$, Deferral logic $\delta$, Prefix $x_{<t}$, Block size $\gamma$
  $\mathbb{T}_\delta(q, p) \doteq (1 - \delta(q, p)) \cdot q + \delta(q, p) \cdot p$
**Output:** `GenSpecSample`$(q, p, \mathbb{T}_\delta, x_{<t}, \gamma)$

---

Contrasting (8) with the deferral risk in (3) for a token-level cascade, the difference is that the cost of deferring to the larger model is *no longer a constant*, but depends on the similarity between $q_t$ and $p_t$, as measured by the total variation (TV) distance between them. Analgous to Lemma 1, we next derive the optimal deferral rule for (8), and then construct a feasible estimator for it.

**Lemma 4** (Optimal deferral for speculative cascades). *The minimizer of (8) is of the form:*

$$r^*(x_{<t}) = 1 \quad \Longleftrightarrow \quad \mathbb{E}_{v \sim \mathbb{P}(\cdot|x_{<t})}\left[\ell(v, q_t)\right] > \mathbb{E}_{v \sim \mathbb{P}(\cdot|x_{<t})}\left[\ell(v, p_t)\right] + \alpha \cdot D_{\text{TV}}(p_t, q_t). \quad (9)$$

When $p_t$ and $q_t$ are similar, the rejection rate for $q_t$ is low, and hence the deferral decision will depend largely on which of the two models yields a lower expected loss. When $p_t$ and $q_t$ are very different, the optimal decision is to defer to $p_t$ only when it yields a substantially lower loss than $q_t$.

*Plug-in estimator for (9).* The optimal rule requires estimating expectations with respect the data generating distribution $\mathbb{P}(\cdot|x_{<t})$. We employ similar plug-in estimators as the ones used with token-level cascades (§3.2). When $\ell = \ell_{0\text{-}1}$, we replace the expected 0-1 loss with (one minus) the maximum probability from the model, giving us:

$$\boxed{\hat{r}_{\text{OPT}}(x_{<t}) = 1 \quad \Longleftrightarrow \quad \max_v q_t(v) < \max_v p_t(v) - \alpha \cdot D_{\text{TV}}(p_t, q_t).} \quad (10)$$

The efficacy of the plug-in estimator depends on how closely the individual models approximate the data generating distribution $\mathbb{P}(\cdot|x_{<t})$; this is formalized by the following regret bound:

**Lemma 5** (Regret bound for $\hat{r}_{\text{OPT}}$). *Suppose $\ell = \ell_{0\text{-}1}$. Denote $\mathbb{P}_t(v) \doteq \mathbb{P}(v|x_{<t})$. Then for fixed $x_{<t}$:*

$$L_{\text{spec}}(\hat{r}_{\text{OPT}}; x_{<t}) - \min_r L_{\text{spec}}(r; x_{<t}) \leq \max_{v \in \mathcal{V}}\left|\mathbb{P}_t(v) - q_t(v)\right| + \max_{v \in \mathcal{V}}\left|\mathbb{P}_t(v) - p_t(v)\right|.$$

One can now run the speculative cascading procedure in Algorithm 5 using (10) as the deferral rule; the corresponding $\delta(\cdot)$ is listed in Table 1. See §D.3 for a similar derivation for $\ell = \ell_{\log}$.

## 5 BEYOND CASCADED DEFERRAL: TOKEN-SPECIFIC INTERLEAVINGS

The deferral rules we have seen so far in (5) and (10) decide between the drafter's distribution $q_t(\cdot)$ and the verifier's distribution $p_t(\cdot)$ by comparing their maximum token probabilities. A downside to

this cascaded form of deferral is that the specific draft token sampled $x_t \sim q_t(\cdot)$ may not be the same as the token that maximizes maximize $q_t(\cdot)$. Thus, even when $x_t$ is of poor quality, we may end up accepting it because $q_t$ happens to be *more peaked* than $p_t$.

**Token-specific interleaving.** To alleviate the above problem, we propose the use of *token-specific deferral rules* $r : \mathcal{V}^{t-1} \times \mathcal{V} \to \{0, 1\}$ that use both the prefix $x_{<t}$ and a candidate token $v$ to provide a binary decision $r(x_{<t}, v) \in \{0, 1\}$, with 0 indicating that the token is of acceptable quality. We may then construct a target distribution of the following form:

$$\pi_{\texttt{Token}}(v) = q_t(v) \cdot (1 - r(x_{<t}, v)) + p_t(v) \cdot \eta, \tag{11}$$

where $\eta = \sum_{v' \in \mathcal{V}} r(x_{<t}, v') \cdot q_t(v')$ is a normalizing term chosen to ensure that $\sum_{v \in \mathcal{V}} \pi_{\texttt{Token}}(v) = 1$. This target distribution closely mimics $q_t(\cdot)$ on tokens that the deferral rule $r$ deems to be of acceptable quality, and defers to $p_t(\cdot)$ otherwise. One can modify the generic speculative sampling algorithm in Algorithm 4 to use $\pi_{\texttt{Token}}$ as the target distribution, as shown in Algorithm 6 in §E.

To design the deferral rule $r$, we propose a heuristic variant of the Diff rule in equation 4 (in §E, we discuss deriving a similar variant of the OPT rule in equation 9). Specifically, we compare the probability that the draft token $v$ is the incorrect next token to $x_{<t}$ according to the data-generating distribution $\mathbb{P}$ with the expected 0-1 loss that we would incur if we were to defer to the verifier $p_t$:

$$r(x_{<t}, v) = 1 \iff 1 - \mathbb{P}(v|x_{<t}) > \mathbb{E}_{v' \sim \mathbb{P}(\cdot|x_{<t})}[\ell_{0\text{-}1}(v', p_t)] + \alpha, \tag{12}$$

for a cost parameter $\alpha$. The following are some simple plug-in approximations to (12), where we approximate $\mathbb{P}(v|x_{<t})$ with either $q_t(v)$ or $p_t(v)$, and the expected 0-1 loss using $\max_{v'} p_t(v')$:

$$\hat{r}_{\texttt{TokenV1}}(x_{<t}, v) = 1 \iff q_t(v) < \max_{v'} p_t(v') - \alpha \tag{13}$$

$$\hat{r}_{\texttt{TokenV2}}(x_{<t}, v) = 1 \iff p_t(v) < \max_{v'} p_t(v') - \alpha \tag{14}$$

$$\hat{r}_{\texttt{TokenV3}}(x_{<t}, v) = 1 \iff p_t(v) < \max_{v'} p_t(v') \cdot (1 - \alpha). \tag{15}$$

where (15) uses a multiplicative plug-in approximation.

The resulting target distributions have an intuitive form. For example, with (13):

$$\pi_{\texttt{TokenV1}}(v) = q_t(v) \cdot \mathbf{1}(v \in \mathcal{T}_\alpha) + p_t(v) \cdot \sum_{v' \notin \mathcal{T}_\alpha} q_t(v'), \tag{16}$$

where $\mathcal{T}_\alpha = \{v \in \mathcal{V} : q_t(v) \geq \max_{v'} p_t(v') - \alpha\}$ is the set of tokens deemed important. For these tokens, $\pi_{\texttt{TokenV1}}$ approximates $q_t(\cdot)$; for the rest, it is a re-scaled version of $p_t(\cdot)$.

**Contrasting with lossy speculative sampling.** Recall that lossy speculative sampling (Tran-Thien, 2023) also seeks to mimic a token-specific interleaving of $q_t$ and $p_t$, given by $\pi_{\text{Lossy}}(v) = \max\{\min\{q_t(v), \frac{p_t(v)}{1-\alpha}\}, \frac{p_t(v)}{\beta}\}$, for trade-off parameters $\alpha, \beta$ (Lemma 2). However, in some settings, this choice of target distribution may severely limit the range of cost-quality trade-offs that can be achieved by varying $\alpha$ and $\beta$. For example, note that $\pi_{\text{Lossy}}(v) = 0$ whenever $p_t(v) = 0$, making the trade-off parameters $\alpha$ and $\beta$ irrelevant for such tokens. This can be particularly problematic when sampling with a small temperature or when applying top-$P$ sampling, where $q_t$ and $p_t$ may not share the same support. In contrast, our proposed approach to token-specific deferral enables a wider range of trade-offs under both temperature and top-$P$ sampling, by computing the deferral rule $r$ in (11) on unscaled versions of $q_t$ and $p_t$, while interleaving between scaled versions of $q_t$ and $p_t$ (see §C).

In fact, in the extreme case of greedy decoding (sampling with temperature $T = 0$), $\pi_{\text{Lossy}}$ simply degenerates to $p_t$. For this special case, Leviathan et al. (2023) propose an alternate lossy version of speculative decoding with a deterministic rejection criterion similar to (15). Interestingly, our proposed TokenV3 approach reduces to this variant when $T \to 0$ (see §C.4). Thus TokenV3 can be seen as a generic deferral rule that is applicable to both greedy and non-greedy decoding.

## 6 EXPERIMENTAL RESULTS

We compare our speculative cascading techniques with both sequential cascades and standard speculative decoding on a range of language benchmarks, including translation, reasoning, coding, QA, etc. We evaluate speculative cascades constructed from both the **T5 v1.1 family** of encoder-decoder models (Raffel et al., 2020), and **Gemma v2** decoder-only models (Team et al., 2024).[1]

---

[1]Illustrative colab notebook with Gemma models available at: `https://github.com/google-research/google-research/tree/master/speculative_cascades`.

Table 2: Reduction in latency ($T = 1, \gamma = 5$) when matching the quality of the large model (cols 2–7), and the best quality metric without exceeding the latency of the large model (cols 8–13). Quality is measured in terms of BLEU for WMT and ROUGE-2 for XSum and CNNDM. Rows 1–4 are the baselines; Rows 5–6 contain the proposed method with old deferral rules (§3); Rows 7–8 are with new deferral rules (§4). See §F.2–F.3 for results with varying temperatures and top-$P$ sampling.

| | Latency↓ when matching large model's quality | | | | | | Best quality *without* exceeding large model's latency | | | | | |
| | Small → Large | | | Small → XL | | | Small → Large | | | Small → XL | | |
| Method | WMT | XSum | CNNDM | WMT | XSum | CNNDM | WMT | XSum | CNNDM | WMT | XSum | CNNDM |
|---|---|---|---|---|---|---|---|---|---|---|---|---|
| SeqCascade [Chow] | 1.55× | 0.84× | 0.98× | 2.46× | 0.93× | 0.94× | 16.56 | 12.97 | 9.91 | 16.29 | 16.40 | 11.18 |
| TokenCascade [Chow] | 1.03× | 0.93× | 1.40× | 1.46× | 0.82× | 1.51× | 16.52 | 13.30 | 10.36 | 16.65 | 17.09 | 11.44 |
| SpecDecode [Lossy] | 1.61× | 1.10× | 1.57× | 2.17× | 1.28× | 2.07× | 17.26 | 13.90 | 10.43 | 16.94 | 17.36 | 11.53 |
| BiLD* | 1.34× | 1.04× | 1.38× | 1.85× | 1.28× | 1.84× | 16.49 | 13.81 | 10.14 | 15.90 | 17.35 | 11.35 |
| SpecCascade [Chow] | 1.43× | 1.04× | 1.41× | 2.01× | 1.28× | 1.97× | 17.76 | 13.82 | 10.28 | 16.35 | 17.36 | 11.39 |
| SpecCascade [Diff] | 1.79× | 1.17× | 1.75× | 2.44× | 1.30× | 2.15× | 18.04 | 14.00 | 10.64 | 18.07 | 17.37 | 11.67 |
| SpecCascade [OPT] | **1.95×** | 1.17× | 1.80× | **2.61×** | 1.34× | **2.21×** | 18.33 | 14.10 | 10.86 | 18.09 | 17.48 | 11.85 |
| SpecCascade [Token] | 1.85× | **1.18×** | **1.89×** | 2.50× | **1.40×** | 1.89× | **22.50** | **15.85** | **12.63** | **22.70** | **18.79** | **12.63** |

**Cascades versus SpecDecode evaluation**. Our evaluation protocol is markedly different from the standard evaluation of speculative decoding algorithms, where the goal is to speed up inference with a large model while preserving its output distribution. In contrast, our focus is on **trading-off quality for lower inference costs by interleaving two models** of different sizes. We also do *not* claim to develop a new state-of-the-art method for fast LM inference. Furthermore, the speculative cascades we design build on the original speculative decoding algorithm Leviathan et al. (2023). While one could potentially also adapt our proposal to other recent variants of speculative decoding (Cai et al., 2024; Li et al., 2024a), these involve a wholly orthogonal suite of techniques to what we propose (such as architectural changes, allowing for multiple drafts, distillation, and so on; see §B).

**Proposed methods and baselines.** We evaluate our proposed speculative cascades with four deferral rules: (i) Chow in (2), (ii) Diff in (5), (iii) OPT in (10), and (iv) the Token-specific rule in (15). Of these, (i) and (ii) are existing deferral rules, while (iii) and (iv) are new rules we propose. We also present results for the V1 and V2 variants of the token-specific rules in §F.7.

We compare these with the following cascading and speculative decoding baselines:

(i) *Sequence-level cascade* (Jitkrittum et al., 2023; Gupta et al., 2024) based on sequence-level Chow's rule in (1) (SeqCascade [Chow]).

(ii) *Token-level cascade* outlined in Algorithm 2, with token-level Chow's rule in (2) used for deferral (Chow, 1970; Gupta et al., 2022) (TokenCascade [Chow]).

(iii) *Lossy speculative decoding* described in §2, with both $\beta = 1$ (Leviathan et al., 2023; Zhou et al., 2024) (SpecDecode [Lossy]) and $\beta$ tuned using the procedure in Tran-Thien (2023) (Lossy*).

(iv) *Big-Little Decoder approach* (Kim et al., 2023), with both the original deterministic rejection rule (BiLD), and the stochastic rejection sampling variant of their method described in §B (BiLD*).

**Fine-tuned T5 cascades.** Our experiments on T5 models are based on the setup in Zhou et al. (2024); see §F.1 for details. We use T5-small (77M) as the small model, and either T5-large (800M) or T5-XL (3B) as the large model. In each case, we supervised fine-tune these models on three tasks: WMT EN→DE translation (Bojar et al., 2014), CNN/DM summarization (Hermann et al., 2015), and XSum abstractive summarization (Narayan et al., 2018). We use temperatures $T = 0, 0.1, 0.5, 1.0$, and block sizes $\gamma = 3, 5, 7$ (full results in §F). Following the protocol in Leviathan et al. (2023); Zhou et al. (2024), to measure latency, we evaluate the wall-clock decoding time with batch size 1.

In Table 2, we report for the each method, (i) the reduction in latency from T5 cascades when matching the quality of the large model, and (ii) the best quality it can deliver without exceeding the latency of the large model. SeqCascade and TokenCascade are often seen to fare poorly on both quality and latency metrics, with the exception of WMT, where SeqCascade yields non-trivial speed-ups. SpecCascade [Token] often yields the highest speed-up and the best quality metrics, with OPT coming in second. The reason the Token-specific rule fares better than OPT and Diff is because the latter compute their deferral decisions based on which of $q_t(\cdot)$ and $p_t(\cdot)$ is more *peaked*; this can be a disadvantage when the sampled token is not close to the distribution mode, which is likely to happen when applying temperature sampling with a high temperature.

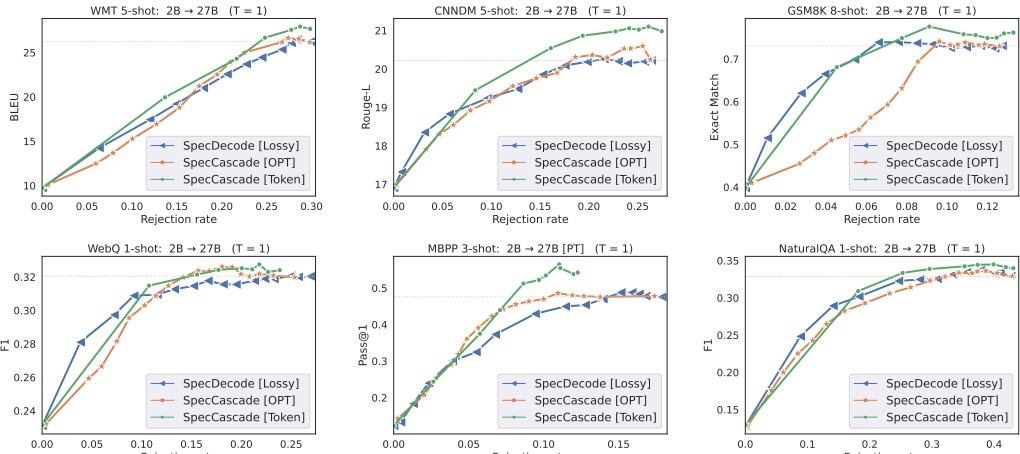

Figure 3: Plots of quality vs. rejection rate for methods that interleave Gemma 2B with Gemma 27B ($\gamma = 1$). We use instruction-tuned models; for MBPP we report additional results with pre-trained models. See §F.7 for remaining plots, comparison to (13–14) and results on 2B → 9B cascades.

We present plots of quality vs. latency for the different methods in Figure 4 in §F. In each case, we vary the trade-off parameter $\alpha$, and plot the quality metric as a function of the relative latency to the large model. While methods that use speculative execution are considerably faster than sequential cascades (`TokenCascade [Chow]`), the latter offer better quality in the low-latency regimes. This is because unlike speculative approaches, which always call the large model after every $\gamma$ steps, sequential cascades invoke the large model only when the small model defers. In §F.5–F.6, we present additional comparisons to `SpecDecode [Lossy⋆]`, and the original `BiLD` algorithm (Kim et al., 2023).

We also report results with varying temperatures in §F.2, and with top-$P$ sampling in §F.3. As the temperature or $P$ becomes smaller, `SpecDecode [Lossy]` yields comparable quality as our methods, but is severely limited in the range of cost-quality trade-offs it offers (see discussion in §5). In contrast, `SpecCascade [Token]` offers a wider range of trade-off points even for low temperature or $P$ values, and is able to match the quality of the larger model at lower latencies (see Table 5).

**Few-shot Gemma cascades.** To evaluate the Gemma model cascades, we use few-shot prompting with 8 language benchmarks: WMT, CNN/DM, GSM8K, MBPP, SQuAD 2.0, WebQuestions, NaturalQA and TriviaQA; many of these feature in the SpecBench suite (Xia et al., 2024). Figure 3 presents plots of quality vs. rejection rate with a 2B drafter and 27B verifier for $\gamma = 1$. For brevity, we only compare the methods that fare the best in the previous experiments. With the exception of TriviaQA, `SpecCascade [Token]` is able to both match the 27B's quality at a lower rejection rate and yield the best overall quality, often better than 27B. Since all three methods use the exact same implementation but with different rejection criteria, we directly compare their rejection rates.

Interestingly, the `OPT` rule is not as effective as it was with the T5 models. We attribute this to the differences in distributions between the two setups. With T5, the maximum token probability served as a good indicator of token accuracy for both $q$ and $p$. With Gemma models, however, we expect the large model to have a closer alignment with the data generating distribution (due to it being several billion parameters apart from the smaller model), and hence using the large model probabilities to measure confidence for both the small and large model (15) yields better trade-offs than comparing the modes from the two model distributions. More generally, we expect `SpecCascade` to yield significant gains over `SpecDecode` when there exists a slice of data where the small model performs comparable to or better than the large model. The larger this slice, the larger is the improvement.

**Conclusions.** We have proposed new speculative cascading techniques that use a combination of auto-regressive drafting and parallel verification to implement their deferral rule, and shown that they yield better cost-quality trade-offs than standard sequential cascades and speculative decoding. A limitation of our approach is that while it offers lower latency via parallel execution, it also incurs a higher total compute cost compared to sequential cascades. In the future, we wish to replace our plug-in estimators with a router model (Gupta et al., 2024) trained on ground-truth samples to approximate the optimal rule, and to extend our proposal to more than two models.

**Acknowledgements.** We thank Ananda Theertha Suresh and Ziteng Sun for insightful discussions and for help implementing speculative cascades with Gemma models.

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

# Appendix

## Table of Contents

# A PROOFS

## A.1 PROOF OF LEMMA 1

*Proof.* Expanding the loss in (3), we have:

$$
\begin{aligned}
& L_{\text{def}}(r; x_{<t}) \\
&= \big(1 - r(x_{<t})\big) \cdot \mathbb{E}_{x_t \sim \mathbb{P}(\cdot|x_{<t})} \left[\ell(x_t, q_t)\right] + r(x_{<t}) \cdot \big(\mathbb{E}_{x_t \sim \mathbb{P}(\cdot|x_{<t})} \left[\ell(x_t, p_t)\right] + \alpha\big) \\
&= r(x_{<t}) \cdot \big(\mathbb{E}_{x_t \sim \mathbb{P}(\cdot|x_{<t})} \left[\ell(x_t, p_t)\right] + \alpha - \mathbb{E}_{x_t \sim \mathbb{P}(\cdot|x_{<t})} \left[\ell(x_t, q_t)\right]\big) + \mathbb{E}_{x_t \sim \mathbb{P}(\cdot|x_{<t})} \left[\ell(x_t, q_t)\right].
\end{aligned}
$$

This objective is minimized by a deferral rule $r : \mathcal{V}^{t-1} \to \{0, 1\}$ that minimizes, for each prefix $x_{<t}$, the term within the parenthesis. Therefore the minimizer $r^*(x_{<t}) = 1$ whenever the term within the parenthesis is negative:

$$
\mathbb{E}_{x_t \sim \mathbb{P}(\cdot|x_{<t})} \left[\ell(x_t, p_t)\right] + \alpha - \mathbb{E}_{x_t \sim \mathbb{P}(\cdot|x_{<t})} \left[\ell(x_t, q_t)\right] < 0,
$$

and $r^*(x_{<t}) = 0$ otherwise. Re-arranging the terms completes the proof. $\qquad\square$

## A.2 PROOF OF LEMMA 2

*Proof.* The proof follows straight-forwardly from the results in (Tran-Thien, 2023). Recall from §2 that the lossy speculative decoding procedure of (Tran-Thien, 2023) accepts a draft token $x$ with probability:

$$
\kappa(x) = \min \left\{1, \frac{p(x)}{(1 - \alpha) \cdot q(x)}\right\}, \tag{17}
$$

and replaces a rejected draft token with a token sampled from the residual distribution:

$$
p_{\text{res}}(x) = \text{norm} \left(\max \left\{0, \frac{1}{\beta} \cdot p(x) - q(x)\right\}\right), \tag{18}
$$

for parameters $\alpha \in [0, 1)$ and $\beta \geq 1 - \alpha$.

We need to show that running Algorithm 4 with the target distribution:

$$
\pi(x) = \max \left\{\min \left\{q(x), \frac{p(x)}{1 - \alpha}\right\}, \frac{p(x)}{\beta}\right\}
$$

results in the same acceptance probability (17) and residual distribution (18).

The acceptance probability for a draft token $x$ when running Algorithm 4 on $\pi$ is given by:

$$
\kappa^\pi(x) = \min \left\{1, \frac{\pi(x)}{q(x)}\right\}.
$$

The corresponding residual distribution is given by:

$$
p_{\text{res}}^\pi(x) = \text{norm} \left(\max \left\{0, \pi(x) - q(x)\right\}\right).
$$

We consider three possible cases:

**Case (i):** $q(x) > \frac{1}{1-\alpha} \cdot p(x) \geq \frac{1}{\beta} \cdot p(x)$. In this case, $\pi(x) = \frac{1}{1-\alpha} \cdot p(x)$. As a result:

$$
\kappa^\pi(x) = \min \left\{1, \frac{p(x)}{(1 - \alpha) \cdot q(x)}\right\} = \kappa(x);
$$

$$
\begin{aligned}
p_{\text{res}}^\pi(x) &= \text{norm} \left(\max \left\{0, \frac{1}{1 - \alpha} \cdot p(x) - q(x)\right\}\right) \\
&= 0 = \text{norm} \left(\max \left\{0, \frac{1}{\beta} \cdot p(x) - q(x)\right\}\right) = p_{\text{res}}(x).
\end{aligned}
$$

**Case (ii):** $\frac{1}{1-\alpha} \cdot p(x) \geq \frac{1}{\beta} \cdot p(x) > q(x)$**.** In this case, $\pi(x) = \frac{1}{\beta} \cdot p(x)$. As a result:

$$\kappa^\pi(x) = \min\left\{1, \frac{p(x)}{\beta \cdot q(x)}\right\} = 1 = \min\left\{1, \frac{p(x)}{(1-\alpha) \cdot q(x)}\right\} = \kappa(x);$$

$$p_{\text{res}}^\pi(x) = \text{norm}\left(\max\left\{0, \frac{1}{\beta} \cdot p(x) - q(x)\right\}\right) = p_{\text{res}}(x).$$

**Case (iii):** $\frac{1}{1-\alpha} \cdot p(x) \geq q(x) \geq \frac{1}{\beta} \cdot p(x)$**.** In this case, $\pi(x) = q(x)$. As a result:

$$\kappa^\pi(x) = 1 = \min\left\{1, \frac{p(x)}{(1-\alpha) \cdot q(x)}\right\} = \kappa(x);$$

$$p_{\text{res}}^\pi(x) = 0 = \text{norm}\left(\max\left\{0, \frac{1}{\beta} \cdot p(x) - q(x)\right\}\right) = p_{\text{res}}(x).$$

In all three cases, the acceptance probabilities and residual distributions are identical. $\qquad\square$

## A.3 PROOF OF LEMMA 3

*Proof.* Under a target distribution $\pi_t$, the probability of a draft token drawn from $q_t$ being is rejected is given by (Leviathan et al., 2023):

$$\text{rejection probability} = \sum_{v \in \mathcal{V}} q_t(v) \cdot \left(1 - \min\left\{1, \frac{\pi_t(v)}{q_t(v)}\right\}\right)$$

$$= 1 - \sum_{v \in \mathcal{V}} \min\{q_t(v), \pi_t(v)\}$$

$$= \sum_{v \in \mathcal{V}} \pi_t(v) - \sum_{v \in \mathcal{V}} \min\{q_t(v), \pi_t(v)\}$$

$$= \sum_{v \in \mathcal{V}} \max\{0, \pi_t(v) - q_t(v)\}.$$

Expanding $\pi_t$, the rejection probability becomes:

$$\text{rejection probability} = \sum_{v \in \mathcal{V}} \max\{0, (1 - r(x_{<t})) \cdot q_t(v) + r(x_{<t}) \cdot p_t(v) - q_t(v)\}$$

When $r(x_{<t}) = 1$, we have:

$$\text{rejection probability} = \sum_{v \in \mathcal{V}} \min\{0, p_t(v) - q_t(v)\} = D_{\text{TV}}(p_t, q_t) = r(x_{<t}) \cdot D_{\text{TV}}(p_t, q_t).$$

When $r(x_{<t}) = 0$, we have:

$$\text{rejection probability} = 0 = r(x_{<t}) \cdot D_{\text{TV}}(p_t, q_t),$$

as desired. $\qquad\square$

## A.4 PROOF OF LEMMA 4

*Proof.* Expanding the deferral risk in (8), we have:

$$L_{\text{spec}}(r; x_{<t}) = r(x_{<t}) \cdot \left(\mathbb{E}_{x_t \sim \mathbb{P}(\cdot|x_{<t})}[\ell(x_t, p_t)] + \alpha \cdot D_{\text{TV}}(p_t, q_t) - \mathbb{E}_{x_t \sim \mathbb{P}(\cdot|x_{<t})}[\ell(x_t, q_t)]\right)$$
$$+ \mathbb{E}_{x_t \sim \mathbb{P}(\cdot|x_{<t})}[\ell(x_t, q_t)].$$

This objective is minimized by a deferral rule $r : \mathcal{V}^{t-1} \to \{0, 1\}$ that minimizes, for each prefix $x_{<t}$, the term within the parenthesis. Therefore the minimizer $r^*(x_{<t}) = 1$ whenever the term within the parenthesis is negative:

$$\mathbb{E}_{x_t \sim \mathbb{P}(\cdot|x_{<t})}[\ell(x_t, p_t)] + \alpha \cdot D_{\text{TV}}(p_t, q_t) - \mathbb{E}_{x_t \sim \mathbb{P}(\cdot|x_{<t})}[\ell(x_t, q_t)] < 0,$$

and $r^*(x_{<t}) = 0$ otherwise. Re-arranging the terms completes the proof. $\qquad\square$

## A.5 PROOF OF LEMMA 5

For a fixed prefix $x_{<t}$, we can write the deferral risk in (8) as:

$$L_{\text{spec}}(r; x_{<t}) = r(x_{<t}) \cdot \left( \mathbb{E}_{x_t \sim \mathbb{P}(\cdot|x_{<t})} [\ell(x_t, p_t)] + \alpha \cdot D_{\text{TV}}(p_t, q_t) - \mathbb{E}_{x_t \sim \mathbb{P}(\cdot|x_{<t})} [\ell(x_t, q_t)] \right) + C,$$

where $C$ is a term independent of the deferral rule $r$. Let $r^* : \mathcal{V}^{t-1} \to \{0, 1\}$ denote the optimal deferral rule that minimizes $L_{\text{spec}}$ for any prefix $x_{<t}$. We then have:

$$L_{\text{spec}}(\hat{r}_{\text{OPT}}; x_{<t}) - L_{\text{spec}}(r^*; x_{<t})$$
$$= (\hat{r}_{\text{OPT}}(x_{<t}) - r^*(x_{<t})) \cdot \left( \mathbb{E}_{x_t \sim \mathbb{P}(\cdot|x_{<t})} [\ell(x_t, p_t)] + \alpha \cdot D_{\text{TV}}(p_t, q_t) - \mathbb{E}_{x_t \sim \mathbb{P}(\cdot|x_{<t})} [\ell(x_t, q_t)] \right).$$

Adding and subtracting $\max_{v \in \mathcal{V}} q_t(v) - \max_{v \in \mathcal{V}} p_t(v)$ to the term within the second parenthesis, we get:

$$L_{\text{spec}}(\hat{r}_{\text{OPT}}; x_{<t}) - L_{\text{spec}}(r^*; x_{<t})$$
$$= (\hat{r}_{\text{OPT}}(x_{<t}) - r^*(x_{<t})) \cdot \left( \max_{v \in \mathcal{V}} q_t(v) + \alpha \cdot D_{\text{TV}}(p_t, q_t) - \max_{v \in \mathcal{V}} p_t(v) \right)$$
$$+ (\hat{r}_{\text{OPT}}(x_{<t}) - r^*(x_{<t})) \cdot \left( \mathbb{E}_{x_t \sim \mathbb{P}(\cdot|x_{<t})} [\ell(x_t, p_t)] - \mathbb{E}_{x_t \sim \mathbb{P}(\cdot|x_{<t})} [\ell(x_t, q_t)] \right.$$
$$\left. - \max_{v \in \mathcal{V}} q_t(v) + \max_{v \in \mathcal{V}} p_t(v) \right)$$
$$= (\hat{r}_{\text{OPT}}(x_{<t}) - r^*(x_{<t})) \cdot \left( \max_{v \in \mathcal{V}} q_t(v) + \alpha \cdot D_{\text{TV}}(p_t, q_t) - \max_{v \in \mathcal{V}} p_t(v) \right)$$
$$+ (\hat{r}_{\text{OPT}}(x_{<t}) - r^*(x_{<t})) \cdot \left( \mathbb{E}_{x_t \sim \mathbb{P}(\cdot|x_{<t})} [\ell(x_t, p_t)] - 1 + \max_{v \in \mathcal{V}} p_t(v) \right)$$
$$+ (\hat{r}_{\text{OPT}}(x_{<t}) - r^*(x_{<t})) \cdot \left( 1 - \max_{v \in \mathcal{V}} q_t(v) - \mathbb{E}_{x_t \sim \mathbb{P}(\cdot|x_{<t})} [\ell(x_t, q_t)] \right)$$
$$\leq (\hat{r}_{\text{OPT}}(x_{<t}) - r^*(x_{<t})) \cdot \left( \max_{v \in \mathcal{V}} q_t(v) + \alpha \cdot D_{\text{TV}}(p_t, q_t) - \max_{v \in \mathcal{V}} p_t(v) \right)$$
$$+ |\hat{r}_{\text{OPT}}(x_{<t}) - r^*(x_{<t})| \cdot \left| \mathbb{E}_{x_t \sim \mathbb{P}(\cdot|x_{<t})} [\ell(x_t, p_t)] - 1 + \max_{v \in \mathcal{V}} p_t(v) \right|$$
$$+ |\hat{r}_{\text{OPT}}(x_{<t}) - r^*(x_{<t})| \cdot \left| 1 - \max_{v \in \mathcal{V}} q_t(v) - \mathbb{E}_{x_t \sim \mathbb{P}(\cdot|x_{<t})} [\ell(x_t, q_t)] \right|$$
$$\leq \underbrace{(\hat{r}_{\text{OPT}}(x_{<t}) - r^*(x_{<t})) \cdot \left( \max_{v \in \mathcal{V}} q_t(v) + \alpha \cdot D_{\text{TV}}(p_t, q_t) - \max_{v \in \mathcal{V}} p_t(v) \right)}_{\text{term}_1}$$
$$+ \underbrace{\left| \mathbb{E}_{x_t \sim \mathbb{P}(\cdot|x_{<t})} [\ell(x_t, p_t)] - 1 + \max_{v \in \mathcal{V}} p_t(v) \right|}_{\text{term}_2} + \underbrace{\left| 1 - \max_{v \in \mathcal{V}} q_t(v) - \mathbb{E}_{x_t \sim \mathbb{P}(\cdot|x_{<t})} [\ell(x_t, q_t)] \right|}_{\text{term}_3}$$

$$(19)$$

where we have used the fact that $|\hat{r}_{\text{OPT}}(x_{<t}) - r^*(x_{<t})| \leq 1$.

We bound each term separately. For the first term, consider two cases: (i) $\max_{v \in \mathcal{V}} q_t(v) + \alpha \cdot D_{\text{TV}}(p_t, q_t) - \max_{v \in \mathcal{V}} p_t(v) \leq 0$ and (ii) $\max_{v \in \mathcal{V}} q_t(v) + \alpha \cdot D_{\text{TV}}(p_t, q_t) - \max_{v \in \mathcal{V}} p_t(v) > 0$. When (i) holds, $\hat{r}_{\text{OPT}}(x_{<t}) = 1$; so irrespective of whether $r^*(x_{<t})$ is 0 or 1,

$$\text{term}_1 \leq \max_{v \in \mathcal{V}} p_t(v) + \alpha \cdot D_{\text{TV}}(p_t, q_t) - \max_{v \in \mathcal{V}} q_t(v) \leq 0$$

When (ii) holds, $\hat{r}_{\text{OPT}}(x_{<t}) = 0$; so irrespective of whether $r^*(x_{<t})$ is 0 or 1,

$$\text{term}_1 \leq - \left( \max_{v \in \mathcal{V}} p_t(v) + \alpha \cdot D_{\text{TV}}(p_t, q_t) - \max_{v \in \mathcal{V}} q_t(v) \right) < 0.$$

Thus we have:

$$\text{term}_1 \leq 0. \tag{20}$$

We next move to the second term. Since $\ell = \ell_{0\text{-}1}$, we have:

$$\begin{aligned}
\text{term}_2 &= \left| \mathbb{E}_{x_t \sim \mathbb{P}(\cdot | x_{<t})} \left[ \ell(x_t, p_t) \right] - 1 + \max_{v \in \mathcal{V}} p_t(v) \right| \\
&= \left| \mathbb{E}_{x_t \sim \mathbb{P}(\cdot | x_{<t})} \left[ 1 \left( x_t \neq \arg\max_{v \in \mathcal{V}} p_t(v) \right) \right] - 1 + \max_{v \in \mathcal{V}} p_t(v) \right| \\
&= \left| \max_{v \in \mathcal{V}} p_t(v) - \sum_{x_t \in \mathcal{V}} \mathbb{P}(x_t | x_{<t}) \cdot 1 \left( x_t = \arg\max_{v \in \mathcal{V}} p_t(v) \right) \right|
\end{aligned}$$

Let $v^* \in \arg\max_{v \in \mathcal{V}} p_t(v)$. Then:

$$\text{term}_2 = |p_t(v^*) - \mathbb{P}(v^* | x_{<t})| \leq \max_{v \in \mathcal{V}} |p_t(v) - \mathbb{P}(v | x_{<t})| . \tag{21}$$

Similarly, we can show that:

$$\text{term}_3 \leq \max_{v \in \mathcal{V}} |q_t(v) - \mathbb{P}(v | x_{<t})| . \tag{22}$$

Substituting (20)–(22) in (19) completes the proof.

| Inference strategy | Deferral decision $\delta(q,p)$ | Target distribution $\pi(u)$ | Execution |
|---|---|---|---|
| SpecDecoding Leviathan et al. (2023) | - | $p(u)$ | Speculative |
| Lossy SpecDecoding (Tran-Thien, 2023) | - | $\max\{\min\{p(u), \frac{q(u)}{1-\alpha}\}, \frac{q(u)}{\beta}\}$ | Speculative |
| BiLD* (Kim et al., 2023) | $\mathbf{1}\big(D(q,p) > \alpha\big)$ | $(1-\delta) \cdot q(u) + \delta \cdot p(u)$ | Speculative |
| Cascade [Chow] (Chow, 1970) | $\mathbf{1}\big(\max_{v \in \mathcal{V}} q(v) < 1 - \alpha\big)$ | $(1-\delta) \cdot q(u) + \delta \cdot p(u)$ | Sequential |
| Cascade [ChowLog] | $\mathbf{1}\big(\text{entropy}(q) > \alpha\big)$ | $(1-\delta) \cdot q(u) + \delta \cdot p(u)$ | Sequential |
| Oracle [Diff] (Jitkrittum et al., 2023) | $\mathbf{1}\big(\max_{v \in \mathcal{V}} q(v) < \max_{v \in \mathcal{V}} p(v) - \alpha\big)$ | $(1-\delta) \cdot q(u) + \delta \cdot p(u)$ | Oracle |
| Oracle [DiffLog] | $\mathbf{1}\big(\text{entropy}(p) < \text{entropy}(q) - \alpha\big)$ | $(1-\delta) \cdot q(u) + \delta \cdot p(u)$ | Oracle |
| SpecCascade [Chow] | $\mathbf{1}\big(\max_{v \in \mathcal{V}} q(v) < 1 - \alpha\big)$ | $(1-\delta) \cdot q(u) + \delta \cdot p(u)$ | Speculative |
| SpecCascade [ChowLog] | $\mathbf{1}\big(\text{entropy}(q) > \alpha\big)$ | $(1-\delta) \cdot q(u) + \delta \cdot p(u)$ | Speculative |
| SpecCascade [Diff01] | $\mathbf{1}\big(\max_{v \in \mathcal{V}} q(v) < \max_{v \in \mathcal{V}} p(v) - \alpha\big)$ | $(1-\delta) \cdot q(u) + \delta \cdot p(u)$ | Speculative |
| SpecCascade [DiffLog] | $\mathbf{1}\big(\text{entropy}(p) < \text{entropy}(q) - \alpha\big)$ | $(1-\delta) \cdot q(u) + \delta \cdot p(u)$ | Speculative |
| SpecCascade [OPT01] | $\mathbf{1}\big(\max_{v \in \mathcal{V}} q(v) < \max_{v \in \mathcal{V}} p(v) - \alpha \cdot D_{\text{TV}}(p,q)\big)$ | $(1-\delta) \cdot q(u) + \delta \cdot p(u)$ | Speculative |
| SpecCascade [OPTLog] | $\mathbf{1}\big(\text{entropy}(p) < \text{entropy}(q) - \alpha \cdot D_{\text{TV}}(p,q)\big)$ | $(1-\delta) \cdot q(u) + \delta \cdot p(u)$ | Speculative |

Table 3: Target distributions associated with different inference algorithms, where $\alpha$ is a free parameter and $\beta \geq 1 - \alpha$ is a parameter dependent on $q, p$ and $\alpha$. The last column indicates whether the execution is sequential (Algorithm 2), via an oracle (Algorithm 3), or speculative (Algorithm 5) (Leviathan et al., 2023). The third row presents a variant of the BiLD algorithm of Kim et al. (2023), where $D(q,p)$ is a measure of discrepancy between $q$ and $p$; the original algorithm differs from (Leviathan et al., 2023) in the use of a deterministic speculative decoding procedure with a dynamic draft window (see §B).

## B  FURTHER RELATED WORK

Several works have studied improving the drafting process in speculative decoding; these include having the drafter and verifier share their backbone (Stern et al., 2018; Kim et al., 2024; Cai et al., 2024; Monea et al., 2023; Hooper et al., 2023; Zhang et al., 2023; Elhoushi et al., 2024; Liu et al., 2024), using multiple small draft models (Chen et al., 2023c; Wang et al., 2024), using tree-structured draft batches (Spector & Re, 2023; Miao et al., 2024), distilling the drafter with the verifier (Zhou et al., 2024), and leveraging multiple sampled drafts (Sun et al., 2024; Chen et al., 2024).

The work that is most closely related to our specific proposal is the Big Little Decoder (BiLD) (Kim et al., 2023), which can be seen as another lossy variant of speculative decoding (Leviathan et al., 2023; Tran-Thien, 2023; Zhou et al., 2024). BiLD has two phases: a *fallback* phase, during which the drafter $q$ is run auto-regressively until its maximum predicted probability is sufficiently low; and a *rollback* phase, during which the verifier $p$ is run in parallel on the prefixes generated by $q$ and rolls back to the point where $D(q,p) > \alpha$, for a metric $D$ that measures discrepancy and threshold $\alpha$. The fallback phase implements Chow's deferral rule in (2), and allows for the draft window size to vary dynamically based on an estimate of how likely the draft tokens will be accepted; the rollback phase can be seen as a deterministic variant of the rejection sampling algorithm of Leviathan et al. (2023).

An advantage of BiLD over the rejection sampling algorithm in (Leviathan et al., 2023) is the use of Chow's rule to vary the draft window size. However, the final target distribution it seeks to mimic, $\mathbb{T}_{\text{BiLD}}(q,p)(v) = \mathbf{1}(D(q,p) \leq \alpha) \cdot q(v) + \mathbf{1}(D(q,p) > \alpha) \cdot p(v)$, is an approximation to $p$; specifically, the target distribution $\pi = \mathbb{T}_{\text{BiLD}}(q,p)$ is chosen to satisfy $D(\pi,p) \leq \alpha$. Hence, in cases where $q$ deviates substantially from $p$, BiLD would choose $p$ as the target distribution, even when $q$ offers better quality on a prefix (where quality can be measured using a suitable loss function). In contrast, our proposed approach in §4 uses speculative decoding to approximate target distributions that seek to *optimally* cascade between $q$ and $p$. In our experiments, we compare the efficacy of using $\mathbb{T}_{\text{BiLD}}$ as the target distribution with the target distributions we propose in this paper (see Table 3).

## C   CONTRASTING SPECULATIVE CASCADES AND LOSSY SPECULATIVE SAMPLING UNDER DIFFERENT SAMPLING SCHEMES

We contrast how speculative cascades and lossy speculative sampling behave under temperature sampling, top-$P$ sampling and greedy decoding.

### C.1   SPECULATIVE CASCADES UNDER TEMPERATURE SAMPLING AND TOP-$P$ SAMPLING

When implementing speculative cascades with temperature sampling and top-$P$ sampling, we compute the deferral rule on the original distributions $p$ and $q$, but use the deferral decisions to interleave the temperature-scaled (or top-$P$ truncated) versions of $p$ and $q$.

For the cascaded deferral rules in Table 1, with the exception of OPT, we construct the target distribution in (6) as follows:

$$\tilde{\pi}_t(v) = (1 - \delta(q_t, p_t)) \cdot \mathbb{S}(q_t)(v) + \delta(q_t, p_t) \cdot \mathbb{S}(p)(v), \tag{23}$$

where $\mathbb{S} : \Delta_\mathcal{V} \to \Delta_\mathcal{V}$ denotes a transformation of the distribution such as temperature scaling or top-$P$ truncation, and $\delta : \Delta_\mathcal{V} \times \Delta_\mathcal{V} \to \{0, 1\}$ denotes the deferral rule. One may run Algorithm 5 with $\tilde{\pi}_t$ as the target distribution, and $\mathbb{S}(q_t)$ and $\mathbb{S}(p_t)$ as the drafter and verifier distributions.

In the case of the OPT rule, we would formulate the constrained problem in (7) to use the TV distance between the distributions $\mathbb{S}(q_t)$ and $\mathbb{S}(p_t)$ to measure the rejection rate. The optimal deferral rule in Lemma 4 would now use $D_{\text{TV}}(\mathbb{S}(p_t), \mathbb{S}(q_t))$ instead of $D_{\text{TV}}(p_t, q_t)$. To construct a plug-in estimator to this optimal rule, we still prescribe using the unscaled probabilities $q_t$ and $p_t$ to estimate the expected loss, giving us, for $\ell = \ell_{0\text{-}1}$:

$$\delta(p_t, q_t) = 1 \quad \iff \quad \max_v q_t(v) \; < \; \max_v p_t(v) - \alpha \cdot D_{\text{TV}}(\mathbb{S}(p_t), \mathbb{S}(q_t)).$$

For the token-specific deferral rules in §5, we compute the target distribution in (16) as follows:

$$\pi_{\texttt{Token}}(v) = \mathbb{S}(q_t)(v) \cdot (1 - r(x_{<t}, v)) + \mathbb{S}(p_t)(v) \cdot \eta,$$

where the deferral rule $r(x_{<t}, v')$ is computed on unscaled distributions $q_t$ and $p_t$, and $\eta = \sum_{v' \in \mathcal{V}} r(x_{<t}, v') \cdot \mathbb{S}(q_t)(v')$ is a normalizing term. For example, for the TokenV3 deferral rule in (15), we compute the target distribution as:

$$\pi(v) = \mathbb{S}(q_t)(v) \cdot \mathbf{1}\big(v \in \mathcal{T}_\alpha\big) \; + \; \mathbb{S}(p_t)(v) \cdot \sum_{v' \notin \mathcal{T}_\alpha} \mathbb{S}(q_t)(v'),$$

where $\mathcal{T}_\alpha = \{v \in \mathcal{V} : p_t(v) \geq \max_{v'} p_t(v') \cdot (1 - \alpha)\}$ is the set of top-ranked tokens by the original (unscaled) distribution $p_t$.

### C.2   CONTRASTING WITH LOSSY SPECULATIVE SAMPLING UNDER TEMPERATURE SAMPLING

When implementing lossy speculative decoding under temperature sampling, following Leviathan et al. (2023); Zhou et al. (2024); Tran-Thien (2023), we compute the acceptance criterion and the residual distribution using temperature-scaled drafter and verifier distributions. Specifically, we accept a draft token $v$ with probability:

$$\min \left\{ 1, \frac{\mathbb{S}(p_t)(v)}{(1 - \alpha) \cdot \mathbb{S}(q_t)(v)} \right\}.$$

Upon rejection, we replace the token by a new token sampled from a residual distribution again constructed from temperature-scaled distributions:

$$\text{norm} \left( \max \left\{ 0, \frac{1}{\beta} \cdot \mathbb{S}(p_t)(\cdot) - \mathbb{S}(q_t)(\cdot) \right\} \right),$$

where $\beta \geq 1 - \alpha$ is a parameter that depends on $\alpha$, $q_t$ and $p_t$, and is such that $\sum_{v \in \mathcal{V}} \pi_{\text{Lossy}}(v) = 1$.

The resulting target distribution that the method samples from takes the form:

$$\pi_{\text{Lossy}}(v) = \max \left\{ \min \left\{ \mathbb{S}(q_t)(v), \frac{\mathbb{S}(p_t)(v)}{1 - \alpha} \right\}, \frac{\mathbb{S}(p_t)(v)}{\beta} \right\}.$$

This choice of target distribution may severely limit the range of cost-quality trade-offs that can be achieved by varying $\alpha$ and $\beta$. For example, observe that

$$\mathbb{S}(p_t)(v) = 0 \implies \pi_{\text{Lossy}}(v) = 0,$$

and so the the trade-off parameters $\alpha$ and $\beta$ are not effective on tokens for which $\mathbb{S}(p_t)(v) = 0$.

This problem is exacerbated when sampling with temperature 0 (i.e., greedy decoding) where $\pi_{\text{Lossy}}(v)$ becomes identical to $p_t$, making $\alpha$ and $\beta$ irrelevant.

**Lemma 6.** *When sampling with temperature 0 (i.e., greedy decoding), $\pi_{\text{Lossy}}(v) = \mathbb{S}_0(p_t)$.*

*Proof.* Applying temperature sampling with temperature 0 to a distribution $p$ is equivalent to sampling from a transformed distribution $\mathbb{S}_0(p)$, where $\mathbb{S}_0(p)$ assigns a probability of 1 to $\arg\max_v p(v)$. When $\mathbb{S}_0(p_t)(v) = 0$, we have that:

$$\pi_{\text{Lossy}}(v) = \max\left\{\min\left\{\mathbb{S}_0(q_t)(v), 0\right\}, 0\right\} = 0.$$

Since $\sum_v \pi_{\text{Lossy}}(v) = 1$, it turns out that: $\pi_{\text{Lossy}}(v) = 1$ whenever $\mathbb{S}_0(p_t)(v) = 1$. Thus: $\pi_{\text{Lossy}}(v) = \mathbb{S}_0(p_t)$. $\qquad\square$

In contrast, because both our cascaded deferral rule and the token-specific deferral rules work with unscaled distributions, they provide meaningful trade-offs under temperature sampling, including when sampling with temperature 0.

### C.3 Contrasting with lossy speculative sampling under top-$P$ sampling

When implementing lossy speculative sampling under top-$P$ sampling, we accept a draft token $v$ with probability:

$$\min\left\{1, \frac{\mathbb{S}_P(p_t)(v)}{(1-\alpha) \cdot \mathbb{S}_P(q_t)(v)}\right\},$$

where $\mathbb{S}_P(p)$ truncates the distribution $p$ to only retain the top-$P$ fraction of tokens (i.e. smallest subset of tokens whose cumulative probability exceeds $P$).

Notice that as $P$ gets smaller, $\mathbb{S}_P(p)$ assigns zero probabilities to a majority of tokens. As a result for most draft token candidates $v$, the above criterion evaluates to 0, and the trade-off parameter $\alpha$ has no effect on those tokens. Hence as $P \to 0$, the trade-off parameter $\alpha$ becomes vacuous, and thus lossy speculative decoding fails to offer meaningful trade-offs. In fact, mirroring Lemma 6, lossy speculative decoding becomes identical to standard *loss-less* speculative decoding when $P \to 0$.

A speculative cascade does not suffer from the same issue as it uses the trade-off parameter $\alpha$ not as a scaling parameter in the acceptance criterion, but to construct a new target distribution that is amenable to a higher acceptance rate even under top-$P$ sampling. For example, with the `TokenV3` deferral rule in (15), a draft token $v$ is accepted with probability:

$$\min\left\{1, \frac{\mathbb{S}_P(\pi_t(v))}{\mathbb{S}_P(q_t(v))}\right\},$$

where $\pi_t$ is a new target distribution defined using the trade-off parameter $\alpha$ that interleaves between $p_t$ and $q_t$ as follows:

$$\pi_t(v) = \mathbb{S}_P(q_t)(v) \cdot \mathbf{1}\left(v \in \mathcal{T}_\alpha\right) + \mathbb{S}_P(p_t)(v) \cdot \sum_{v' \notin \mathcal{T}_\alpha} \mathbb{S}_P(q_t)(v'),$$

where $\mathcal{T}_\alpha = \{v \in \mathcal{V} : p_t(v) \geq \max_{v'} p_t(v') \cdot (1-\alpha)\}$ is the set of top ranked tokens by the original untruncated distribution $p_t(\cdot)$. Since the top-ranked tokens $\mathcal{T}_\alpha$ are computed using the untruncated distribution, varying the trade-off parameter $\alpha$ still produces meaningful cost-quality trade-offs when speculatively sampling from $\pi_t(v)$.

Table 4: Acceptance criterion for different speculative inference strategies under non-greedy and greedy decoding. Rows 2 and 3 indicates that under 0 temperature, speculative cascade with the `TokenV3` deferral rule has an identical acceptance criterion as `SpecDecode [Lossy, Greedy]`; see Lemma 7.

| Method | Ref. | Acceptance Criterion | |
|---|---|---|---|
| | | $T > 0$ | $T = 0$ |
| `SpecDecode [Lossy]` | Leviathan et al. (2023) | $\min\left\{1, \frac{\mathbb{S}(p(v))}{(1-\alpha)\cdot\mathbb{S}(q(v))}\right\}$ | $\mathbf{1}(v = \arg\max_{v'} p(v'))$ |
| `SpecDecode [Lossy, Greedy]` | Leviathan et al. (2023) | - | $p(v) \geq (1-\alpha)\cdot\max_{v'} p(v')$ |
| `SpecCascade [TokenV3]` | This paper | $\min\left\{1, \frac{\mathbb{S}(\pi(v))}{\mathbb{S}(q(v))}\right\}$, where $\pi$ is in (16) | $p(v) \geq (1-\alpha)\cdot\max_{v'} p(v')$ |

## C.4 Lossy speculative greedy decoding variant by Leviathan et al. (2023)

For the special case of greedy decoding, Leviathan et al. (2023) propose an alternate lossy variant of speculative decoding (Appendix A.5 in their paper), where a draft token $v$ is accepted deterministically whenever $p_t(v) \geq (1-\alpha)\cdot\max_{v'} p_t(v')$; when the token is rejected, it is replaced with a new token sampled from $p_t(\cdot)$. We will refer to this variant as `SpecDecode [Lossy, Greedy]`.

We now show that the proposed speculative cascades with the `TokenV3` deferral rule (15) is identical to `SpecDecode [Lossy, Greedy]` when sampling with temperature 0.

**Lemma 7.** *For any fixed trade-off parameter $\alpha \in [0, 1]$,* `SpecCascade [TokenV3]` *is identical to* `SpecDecode [Lossy, Greedy]` *when sampling with temperature 0.*

*Proof.* Let $\mathbb{S}_0(p)$ denote a temperature-scaled one-hot version of distribution $p$ which places all its mass on the mode of $p$. Let $\tilde{p}_t = \mathbb{S}_0(p_t)$ and $\tilde{q}_t = \mathbb{S}_0(q_t)$. With the `TokenV3` rule, the acceptance criterion is computed against the target distribution in (16) with trade-off parameter $\alpha$.

$$\pi_t(v) = \tilde{q}_t(v) \cdot \mathbf{1}\big(v \in \mathcal{T}_\alpha\big) + \tilde{p}_t(v) \cdot \sum_{v' \notin \mathcal{T}_\alpha} \tilde{q}_t(v'),$$

where $\mathcal{T}_\alpha = \{v \in \mathcal{V} : p_t(v) \geq \max_{v'} p_t(v') \cdot (1-\alpha)\}$ is the set of top ranked tokens by the original (unscaled) distribution $p_t(\cdot)$.

Under greedy decoding, the draft token is given by $v^* = \arg\max_{v'} q_t(v)$. We consider two cases: (i) $v^* \in \mathcal{T}_\alpha$ and (ii) $v^* \notin \mathcal{T}_\alpha$.

In the first case, we have $\pi_t(v^*) = \tilde{q}_t(v^*)$. As a result, a draft token $v$ is accepted with probability:

$$\min\left\{1, \frac{\mathbb{S}_0(\tilde{q}_t(v^*))}{\mathbb{S}_0(q_t(v^*))}\right\} = \min\left\{1, \frac{\tilde{q}_t(v^*)}{\tilde{q}_t(v^*)}\right\} = 1.$$

In the second case, it is clear that the draft token $v^*$ is not the maximizer of $p_t(\cdot)$. Furthermore, $\pi_t(v^*) = \tilde{p}_t(v^*)$. As a result, the draft token $v^*$ is rejected since the acceptance probability for the token becomes:

$$\min\left\{1, \frac{\mathbb{S}_0(\tilde{p}_t(v^*))}{\mathbb{S}_0(q_t(v^*))}\right\} = \min\left\{1, \frac{\tilde{p}_t(v^*)}{\tilde{q}_t(v^*)}\right\} = \min\left\{1, \frac{0}{\tilde{q}_t(v^*)}\right\} = 0.$$

It is then replaced with a token sampled from:

$$\text{norm}\left(\max\{0, \mathbb{S}_0(\tilde{p}_t(\cdot)) - \mathbb{S}_0(q_t(\cdot))\}\right) = \text{norm}\left(\max\{0, \tilde{p}_t(\cdot) - \tilde{q}_t(\cdot)\}\right) = \text{norm}\left(\tilde{p}_t(\cdot)\right) = \tilde{p}_t(\cdot),$$

which would produce the token maximizing $p_t(\cdot)$.

In both cases, the sampling procedure is identical to that of `SpecDecode [Lossy, Greedy]`. □

Table 4 summarizes the acceptance criteria for different speculative inference strategies under temperature sampling and what they reduce to under greedy decoding.

## D  OPTIMAL DEFERRAL: ADDITIONAL DISCUSSION

We provide additional discussion for the deferral rules derived in §3 and §4.

### D.1  DERIVATION OF CHOW'S RULE

We show below that Chow's rule is a plug-in estimator to the optimal solution to the following objective

$$L_{\mathrm{rej}}(r; x_{<t}) = \mathbb{E}_{x_t \sim \mathbb{P}(\cdot | x_{<t})}\Big[\big(1 - r(x_{<t})\big) \cdot \ell(x_t, q_t) + r(x_{<t}) \cdot \alpha\Big], \tag{24}$$

where the deferral rule is penalized with a constant penalty $\alpha \in [0, 1]$ for choosing to defer to the large model.

Following the same steps as Lemma 1, it is easy to show:

**Lemma 8.** *The minimizer of (24) is of the form:*

$$r^*(x_{<t}) = 1 \quad \Longleftrightarrow \quad \mathbb{E}_{x_t \sim \mathbb{P}(\cdot | x_{<t})}\left[\ell(x_t, q_t)\right] > \alpha. \tag{25}$$

If $\ell = \ell_{0\text{-}1}$, one may employ a plug-in estimator to (25) by replacing the expected 0-1 loss on $q_t$ with $1 - \max_{v \in \mathcal{V}} q_t(v)$, giving us $\hat{r}_{\mathtt{Chow}}(x_{<t})$ in (2). If $\ell = \ell_{\log}$, one may replace the expected log loss on $q_t$ with the entropy of $q_t$, giving us:

$$\hat{r}_{\mathtt{ChowLog}}(x_{<t}) = 1 \quad \Longleftrightarrow \quad \mathrm{entropy}\big(q(\cdot | x_{<t})\big) > \alpha, \tag{26}$$

where $\mathrm{entropy}(q) = -\sum_{v \in \mathcal{V}} q(v) \cdot \log(q(v))$.

### D.2  OPTIMAL SEQUENTIAL DEFERRAL WHEN $\ell = \ell_{\log}$

Recall that the optimal deferral rule for a sequential cascade in Lemma 1 takes the form:

$$r^*(x_{<t}) = 1 \quad \Longleftrightarrow \quad \mathbb{E}_{x_t \sim \mathbb{P}(\cdot | x_{<t})}\left[\ell(x_t, q_t)\right] > \mathbb{E}_{x_t \sim \mathbb{P}(\cdot | x_{<t})}\left[\ell(x_t, p_t)\right] + \alpha.$$

When $\ell = \ell_{\log}$, we may use the entropy $-\sum_{v \in \mathcal{V}} q_t(v) \cdot \log(q_t(v))$ from $q_t$ as an estimate of its expected log-loss, and similarly for $p_t$, giving us the plug-in estimator:

$$\hat{r}_{\mathtt{DiffLog}}(x_{<t}) = 1 \quad \Longleftrightarrow \quad \sum_{v \in \mathcal{V}} q_t(v) \cdot \log(q_t(v)) < \sum_{v \in \mathcal{V}} p_t(v) \cdot \log(p_t(v)) - \alpha. \tag{27}$$

### D.3  OPTIMAL SPECULATIVE DEFERRAL WHEN $\ell = \ell_{\log}$

Recall that the optimal deferral rule for a speculative cascade in Lemma 4 takes the form:

$$r^*(x_{<t}) = 1 \quad \Longleftrightarrow \quad \mathbb{E}_{x_t \sim \mathbb{P}(\cdot | x_{<t})}\left[\ell(x_t, q_t)\right] > \mathbb{E}_{x_t \sim \mathbb{P}(\cdot | x_{<t})}\left[\ell(x_t, p_t)\right] + \alpha \cdot D_{\mathrm{TV}}(p_t, q_t).$$

When $\ell = \ell_{\log}$, one may construct a *plug-in* estimator for the above rule by replacing the expected log loss with the entropy from the distribution:

$$\hat{r}_{\mathtt{OPTLog}}(x_{<t}) = 1 \quad \Longleftrightarrow \quad \sum_{v \in \mathcal{V}} q_t(v) \cdot \log(q_t(v)) < \sum_{v \in \mathcal{V}} p_t(v) \cdot \log(p_t(v)) - \alpha \cdot D_{\mathrm{TV}}(p_t, q_t). \tag{28}$$

**Lemma 9** (Regret bound for $\hat{r}_{\mathtt{OPTLog}}$)**.** *Suppose $\ell = \ell_{\log}$. Suppose for a fixed $x_{<t}$, $|\log(q_t(v))| \le B_q$ and $|\log(p_t(v))| \le B_p$, $\forall v \in \mathcal{V}$, for some $B_q, B_p > 0$. Then:*

$$L_{\mathrm{spec}}(r_{\mathtt{OPT}}; x_{<t}) - \min_r L_{\mathrm{spec}}(r; x_{<t}) \le B_q \cdot \sum_{v \in \mathcal{V}} \big|\mathbb{P}(v|x_{<t}) - q_t(v)\big| + B_p \cdot \sum_{v \in \mathcal{V}} \big|\mathbb{P}(v|x_{<t}) - p_t(v)\big|.$$

*Proof.* The proof follows similar steps to that for Lemma 5, except in bounding the resulting term$_2$ and term$_3$ for the log loss. In this case,

$$\mathrm{term}_2 = \left|\mathbb{E}_{x_t \sim \mathbb{P}(\cdot | x_{<t})}\left[\log(p_t(x_t))\right] - \sum_{v \in \mathcal{V}} p_t(v) \cdot \log(p_t(v))\right|$$

$$= \left| \sum_{v \in \mathcal{V}} \mathbb{P}(v|x_{<t}) \cdot \log(p_t(v)) - \sum_{v \in \mathcal{V}} p_t(v) \cdot \log(p_t(v)) \right|$$

$$\leq \sum_{v \in \mathcal{V}} \left| \mathbb{P}(v|x_{<t}) - \sum_{v \in \mathcal{V}} p_t(v) \right| \cdot \log(p_t(v))$$

$$\leq B_p \cdot \sum_{v \in \mathcal{V}} \left| \mathbb{P}(v|x_{<t}) - \sum_{v \in \mathcal{V}} p_t(v) \right|.$$

Similarly,

$$\text{term}_3 \leq \sum_{v \in \mathcal{V}} \left| \mathbb{P}(v|x_{<t}) - \sum_{v \in \mathcal{V}} q_t(v) \right| \cdot \log(q_t(v))$$

$$\leq B_q \cdot \sum_{v \in \mathcal{V}} \left| \mathbb{P}(v|x_{<t}) - \sum_{v \in \mathcal{V}} q_t(v) \right|.$$

Plugging these bounds into the equivalent of (19) in Lemma 5 for the log-loss completes the proof. $\square$

### D.4 OPTIMAL SPECULATIVE DEFERRAL FOR GREEDY DECODING

When applying speculative cascades with greedy decoding, we shall see that both the optimal deferral rule OPT (10) is equivalent to the Diff deferral rule (5).

As detailed in §C.1, when implementing a speculative cascade with temperature-scaled distributions $\tilde{q}_t(v) \propto q_t(v)^{1/T}$ and $\tilde{p}_t(v) \propto p_t(v)^{1/T}$ respectively, for a temperature parameter $T > 0$, the Diff and OPT deferral rules are computed as:

$$\hat{r}_{\text{Diff}}(x_{<t}) = 1 \iff \max_v q_t(v) < \max_v p_t(v) - \alpha,$$

and

$$\tilde{r}_{\text{OPT}}(x_{<t}) = 1 \iff \max_v q_t(v) < \max_v p_t(v) - \alpha \cdot D_{\text{TV}}(\tilde{p}_t, \tilde{q}_t).$$

One may run Algorithm 5 with either $r_{\text{Diff}}$ or $\hat{r}_{\text{OPT}}$ as the deferral rule, and the temperature-scaled $\tilde{q}_t$ as the drafter distribution and $\tilde{p}_t$ as the verifier distribution.

**Lemma 10.** *When $T \to 0$, running Algorithm 5 with $\tilde{r}_{\text{OPT}}$ as the deferral rule and $\tilde{q}_t$ and $\tilde{p}_t$ as the drafter and verifier distributions, is equivalent to running it with $\hat{r}_{\text{Diff}}$ as the deferral rule and $\tilde{q}_t$ and $\tilde{p}_t$ as the drafter and verifier distributions.*

*Proof.* When $T \to 0$, note that $\tilde{q}_t$ and $\tilde{p}_t$ are one-hot encodings of $\arg\max_{v \in \mathcal{V}} q_t(v)$ and $\arg\max_{v \in \mathcal{V}} p_t(v)$ respectively. As a result,

$$D_{\text{TV}}(\tilde{q}_t, \tilde{p}_t) = \mathbf{1}\left( \arg\max_{v \in \mathcal{V}} q_t(v) \neq \arg\max_{v \in \mathcal{V}} p_t(v) \right).$$

When running Algorithm 5 with $\tilde{r}_{\text{OPT}}$ as the deferral rule, we will accept a draft token $v$ with probability:

$$\kappa_{\text{OPT}}(v) = \min\left\{ 1, \frac{(1 - \delta_{\text{OPT}}(q_t, p_t)) \cdot \tilde{q}_t(v) + \delta_{\text{OPT}}(q_t, p_t) \cdot \tilde{p}_t(v)}{\tilde{q}_t(v)} \right\},$$

where

$$\delta_{\text{OPT}}(q, p) = \mathbf{1}\left( \max_{v \in \mathcal{V}} q(v) < \max_{v \in \mathcal{V}} p(v) - \alpha \cdot \mathbf{1}\left( \arg\max_{v \in \mathcal{V}} q(v) \neq \arg\max_{v \in \mathcal{V}} p(v) \right) \right).$$

When running Algorithm 5 with $\tilde{r}_{\text{Diff}}$ as the deferral rule, we will accept a draft token $v$ with probability:

$$\kappa_{\text{Diff}}(v) = \min\left\{ 1, \frac{(1 - \delta_{\text{Diff}}(q_t, p_t)) \cdot \tilde{q}_t(v) + \delta_{\text{Diff}}(q_t, p_t) \cdot \tilde{p}_t(v)}{\tilde{q}_t(v)} \right\},$$

where

$$\delta_{\mathrm{Diff}}(q, p) = \mathbf{1}\left(\max_{v \in \mathcal{V}} q(v) < \max_{v \in \mathcal{V}} p(v) - \alpha\right).$$

We consider two cases:

(i) If $\arg\max_{v \in \mathcal{V}} q_t(v) = \arg\max_{v \in \mathcal{V}} p_t(v)$, then $\tilde{q}_t = \tilde{p}_t$, and irrespective of the outcome of $\delta_{\mathrm{OPT}}(q_t, p_t)$ or $\delta_{\mathrm{Diff}}(q_t, p_t)$, we have that $\kappa_{\mathrm{OPT}}(v) = \kappa_{\mathrm{Diff}}(v)$. Furthermore, the token gets accepted in both cases.

(ii) If $\arg\max_{v \in \mathcal{V}} q_t(v) \neq \arg\max_{v \in \mathcal{V}} p_t(v)$, then

$$\kappa_{\mathrm{OPT}}(v) = 1 - \delta_{\mathrm{OPT}}(q_t, p_t) = \mathbf{1}\left(\max_{v \in \mathcal{V}} q(v) \geq \max_{v \in \mathcal{V}} p(v) - \alpha\right) = 1 - \delta_{\mathrm{Diff}}(q, p) = \kappa_{\mathrm{Diff}}(v).$$

In this case, when the draft token gets rejected, with deferral rule $\tilde{r}_{\mathrm{OPT}}$, we will sample a new token from the residual distribution:

$$\begin{aligned} p_{\mathrm{OPT}}(v) &\propto \min\{0, (1 - \delta_{\mathrm{OPT}}(q_t, p_t)) \cdot \tilde{q}_t(v) + \delta_{\mathrm{OPT}}(q_t, p_t) \cdot \tilde{p}_t(v) - \tilde{q}_t(v)\} \\ &= \delta_{\mathrm{OPT}}(q_t, p_t) \cdot \min\{0, \tilde{p}_t(v) - \tilde{q}_t(v)\} \end{aligned}$$

When a token gets rejected with deferral rule $\tilde{r}_{\mathrm{Diff}}$, we will sample a new token from the residual distribution:

$$p_{\mathrm{Diff}}(v) \propto \delta_{\mathrm{Diff}}(q_t, p_t) \cdot \min\{0, \tilde{p}_t(v) - \tilde{q}_t(v)\}.$$

Since $\arg\max_{v \in \mathcal{V}} q_t(v) \neq \arg\max_{v \in \mathcal{V}} p_t(v)$,

$$p_{\mathrm{OPT}}(v) \propto \delta_{\mathrm{OPT}}(q_t, p_t) \cdot \min\{0, \tilde{p}_t(v) - \tilde{q}_t(v)\} = \delta_{\mathrm{Diff}}(q_t, p_t) \cdot \min\{0, \tilde{p}_t(v) - \tilde{q}_t(v)\} \propto p_{\mathrm{Diff}}(v).$$

Thus both the acceptance probability and the residual distributions for $\hat{r}_{\mathtt{OPT}}$ are the same as the one we would have used had we run Algorithm 5 with $\hat{r}_{\mathtt{Diff}}$ as the deferral rule. $\square$

## D.5  EQUIVALENCE BETWEEN (7) AND (8)

Since the prefix $x_{<t}$ is fixed in (7), the constrained optimization we seek to solve is of essentially of the following form:

$$\min_{r \in \{0,1\}} (1 - r) \cdot c_0 + r \cdot c_1 \quad \text{s.t.} \quad r \cdot c_2 \leq B,$$

for some coefficients $c_0, c_1, c_2 > 0$. Since $r$ is a binary variable, we may formulate an equivalent unconstrained problem with the same minimizer:

$$\min_{r \in \{0,1\}} (1 - r) \cdot c_0 + r \cdot c_1 + \alpha \cdot r \cdot c_2,$$

where we choose $\alpha = 0$ when $c_2 \leq B$ and choose an $\alpha > \frac{1}{c_2} \cdot (c_0 - c_1)$ otherwise. This unconstrained optimization problem is of the form in (8).

# E   TOKEN-SPECIFIC SPECULATIVE CASCADE

We provide a modification of Algorithm 5 to accommodate the token-specific deferral rules in §5.

---

**Algorithm 6** TokenSpecCascade

---

**Input:** Models $q, p$, Token-specific deferral rule $r$, Prefix $x_{<t}$, Block size $\gamma$
  $\mathbb{T}_{\texttt{Token}}(q, p)(v) \doteq q(v) \cdot (1 - r(x_{<t}, v)) + p(v) \cdot \sum_{v' \in \mathcal{V}} r(x_{<t}, v') \cdot q(v')$
**Output:** GenSpecSample$(q, p, \mathbb{T}_{\texttt{Token}}, x_{<t}, \gamma)$

---

**Optimal token-specific deferral.** Similar to §4.3, we may consider deriving the optimal token-specific deferral rule. We start by formulating a similar optimization objective as §4.3. For a fixed prefix $x_{<t}$, this would look like:

$$\min_{r} \; \mathbb{E}_{v \sim \mathbb{P}(\cdot | x_{<t})} \Big[ \ell(v, \pi_{\texttt{Token}})) \Big] \tag{29}$$
$$\text{s.t.} \; D_{\text{TV}}(\pi_{\texttt{Token}}, q_t) \leq B,$$

where $\pi_{\texttt{Token}}(v) \doteq (1 - r(x_{<t}, v)) \cdot q_t(v) + \eta \cdot p_t(v)$ is the target distribution resulting from the choice of $r$, $\eta = \sum_{v' \in \mathcal{V}} r(x_{<t}, v') \cdot q_t(v')$ is a normalization term, and $B > 0$ is a budget parameter.

However, unlike §4.3, the above constrained optimization problem does not directly lend itself to a simple closed-form solution. In some highly simplistic special cases, we may be able to derive a trivial solution. For example, suppose $\ell = \ell_{\text{0-1}}$, and the mode of $q_t$ coincides with that of $\mathbb{P}(\cdot | x_{<t})$, i.e., $\arg\max_{v \in \mathcal{V}} q_t(v) = \arg\max_{v \in \mathcal{V}} \mathbb{P}(v | x_{<t})$; then the optimal token-specific rule is given by $r(x_{<t}, v) = 0$, for all $v \in \mathcal{V}$.

Under more realistic cases, we may not be able to derive a solution as simple as the OPT rule in (10). Therefore, in our experiments, we employ the three heuristic rules in equations 13–15, which are motivated by the form of the simpler Diff rule in (5).

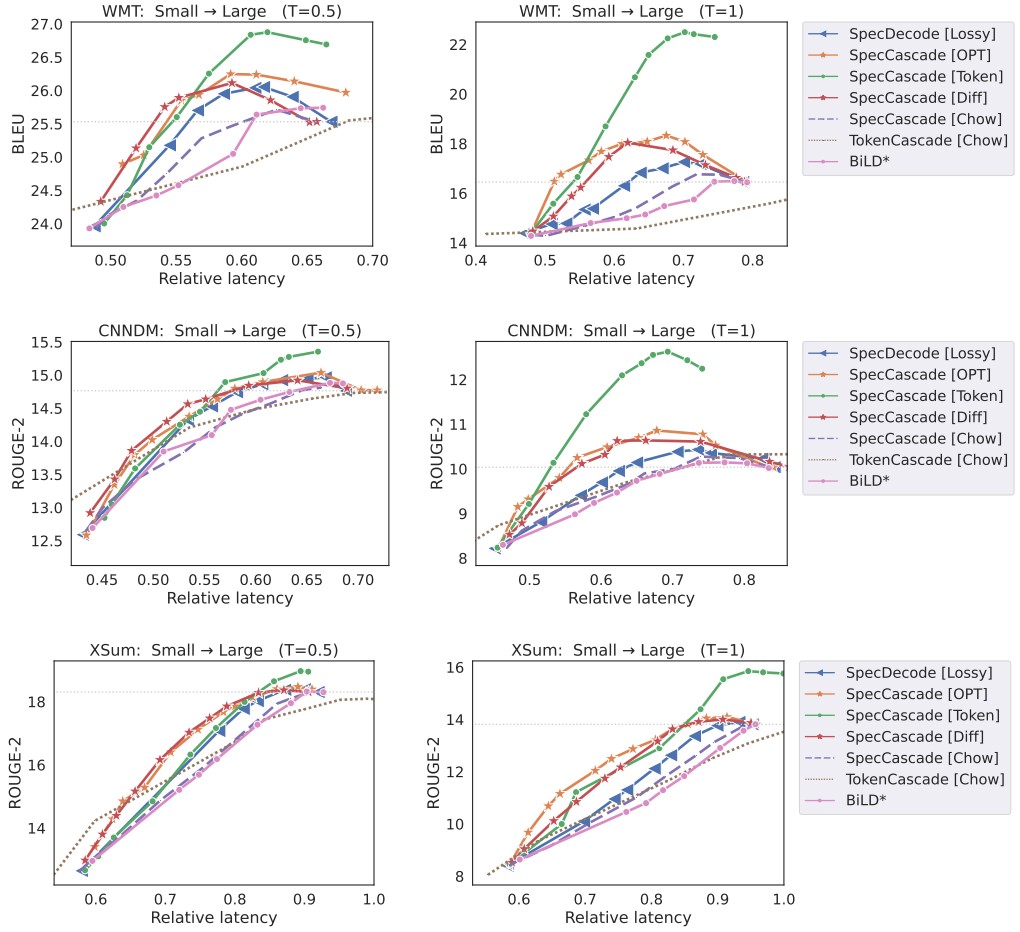

Figure 4: Plots of quality vs. latency for T5 models with temperatures $T = 0.5$ and $T = 1$, and block size $\gamma = 5$. Each method interleaves T5-small with T5-large. We include speculative cascades with the `Chow`, `Diff`, `OPT` and `TokenV3` (referred to as `Token`) deferral rules, and compare it with three baselines: `SpecDecode [Lossy]`, `TokenCascade [Chow]` and `BiLD*`. The $x$-axis tracks the latency *relative* to that of calling the large model on all inputs. The horizontal dotted line denotes the quality of the large model.

## F  ADDITIONAL EXPERIMENTAL DETAILS

We provide additional details about our experimental setup and additional experimental results.

### F.1  EXPERIMENTAL SETUP AND HYPER-PARAMETERS

We first elaborate on our experimental setup and the hyper-parameters used.

**T5 datasets.** For the WMT English to German translation task (Bojar et al., 2014), we use a validation sample of size 3,000 provided with the dataset. We set the maximum input length to 80 and the maximum output length to 80. For the Extreme Summarization (XSum) task (Narayan et al., 2018), we use a validation sample of size 11,305, and set the maximum input length to 1,024 and the maximum output length to 64. For the CNN/Daily Mail summarization task (Hermann et al., 2015), we use a validation sample of size 13368, and set the maximum input length to 2,048 and the maximum output length to 128. Following (Zhou et al., 2024), we use ROUGE-2 as the evaluation metric for the summarization tasks.

We note that Kim et al. (2023) report ROUGE-L metrics for CNN/DM, which generally tend to evaluate to higher values than ROUGE-2. Furthermore, most of their experimental results are with

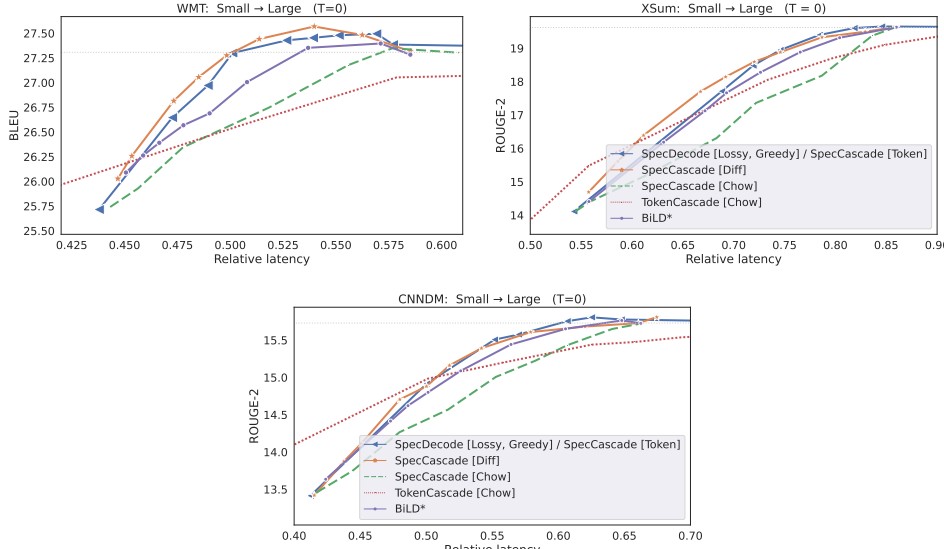

Figure 5: Plots of quality vs. latency for T5 models with **greedy decoding** ($T = 0$) with block size $\gamma = 5$. Each method interleaves T5-small with T5-large. The $x$-axis tracks the latency *relative* to that of calling the large model on all inputs. The horizontal dotted line denotes the quality of the large model. `SpecDecode [Lossy, Greedy]` is the greedy version of lossy speculative decoding proposed in Leviathan et al. (2023). `SpecCascade [Token]` uses the `TokenV3` deferral rule in (15). As noted in §C.4, when $T \to 0$, `SpecDecode [TokenV3]` is identical to `SpecDecode [Lossy, Greedy]`.

greedy decoding ($T = 0$), and hence, the ROUGE-L evaluation metrics they report in their paper tend to be higher for the same T5 models when compared to our numbers for ROUGE-2 with temperature sampling.

**Gemma datasets.** In addition to the WMT EN→DE translation and the CNN/DM summarization datasets, we use the GSM8K (Cobbe et al., 2021) math reasoning dataset, the MBPP (Austin et al., 2021) Python programming dataset, and four question-answering datasets: Natural Questions (Kwiatkowski et al., 2019), TriviaQA (Joshi et al., 2017), WebQuestions (Berant et al., 2013) and the Stanford Question-Answering Dataset (SQuAD) 2.0 (Rajpurkar et al., 2016). In each case, we sample 1,000 prompts for evaluation. We employ few-shot inference, and set the maximum output length to 80 for WMT, to 128 for CNN/DM, to 320 for GSM8K and MBPP, and to 5 for all the question-answering datasets.

**Models.** We construct cascades from T5 v1.1 family of encoder-decoder models (Raffel et al., 2020), of different sizes T5-small (77M), T5-base (250M), T5-large (800M) and T5-XL (3B).[2] We follow the protocol in (Zhou et al., 2024): we initialize with the public checkpoints, pre-train them further for 100K steps, and supervise **finetune** the pre-trained models on the three respective tasks. We finetune them for a maximum of 250K steps on WMT, a maximum of 100K steps on XSum and a maximum of 200K steps on CNNDM.

We construct the Gemma cascades from *instruction-tuned* decoder-only v2 models. For MBPP alone we additionally experiment with *pre-trained* models. We use a 2B drafter, and either a 9B verifier or a 27B verifier (Team et al., 2024).

**Run-time evaluation.** For each dataset, we evaluate the quality metrics on the entire validation set. For the run-time analysis in the T5 experiments, we adopt the protocol followed in Leviathan et al. (2023); Zhou et al. (2024). We randomly sample 500 examples from the validation set, and calculate the wall-clock time taken for decoding with a batch size of 1. We repeat this for three trials and report the average running time. All methods are run on the same TPUv4 device.

---

[2]The pre-trained checkpoints we use are available here.

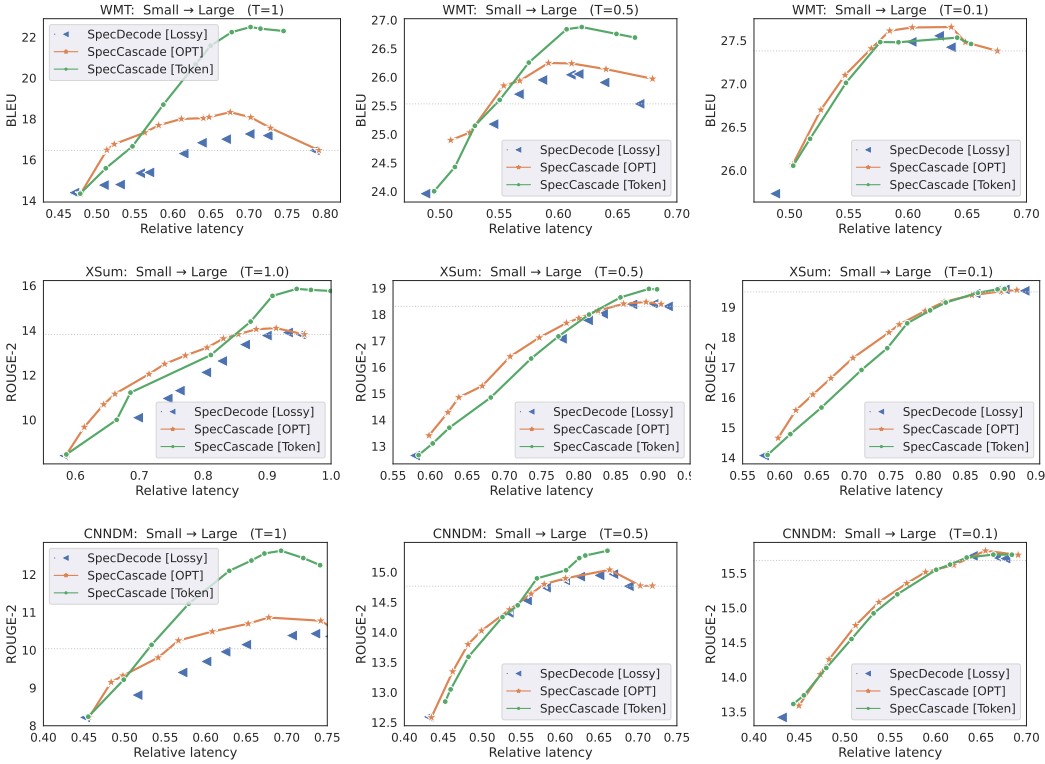

Figure 6: Plots of quality vs. latency for T5 models with **varying temperatures**. Each method interleaves T5-small with T5-large. The $x$-axis tracks the latency *relative* to that of calling the large model on all inputs. The horizontal dotted line denotes the quality of the large model. As the temperature decreases, `SpecDecode [Lossy]` produces fewer unique trade-off points as its acceptance criterion is less susceptible to changes in the trade-off parameter $\alpha$. In contrast, `SpecCascade [Token]`, which uses the `TokenV3` deferral rule, offers a wider range of trade-off points; here the trade-off parameter is not used to construct a traded-off target distribution, and does not feature as a scaling term in the acceptance criterion.

**T5 hyper-parameters.** For the T5 experiments, unless otherwise specified, we set the block-size $\gamma$ to 5 for all methods that use speculative execution. For the token-level cascades, we allow the small model to predict for a maximum of 10 tokens (similar to (Kim et al., 2023)), before invoking the large model. This was needed, as otherwise, the small model would predict a long sequence, and when it eventually defers to the large model, the large model is bottle-necked by the pre-filling of the long prefix accumulated by the small model. We vary the trade-off parameter $\alpha$ to vary the latency and plot quality as a function of latency.

**Gemma inference.** When implementing speculative cascades and speculative decoding with Gemma models, we use block-size $\gamma = 1$. In this case, for each prefix $x_{<t}$, we have the drafter generate one draft token $x_t$ for the next step. We then invoke the verifier with the same prefix, and either accept the draft token $x_t$, or reject and replace it with the verifier's prediction. We repeat this process to generate the entire response.

**BiLD baseline.** For the BiLD method, we adopt the same discrepancy metric $D$ as (Kim et al., 2023) for greedy decoding:

$$D(q, p) = -\log\left(p\left(\underset{v \in \mathcal{V}}{\arg\max}\, q(v)\right)\right),$$

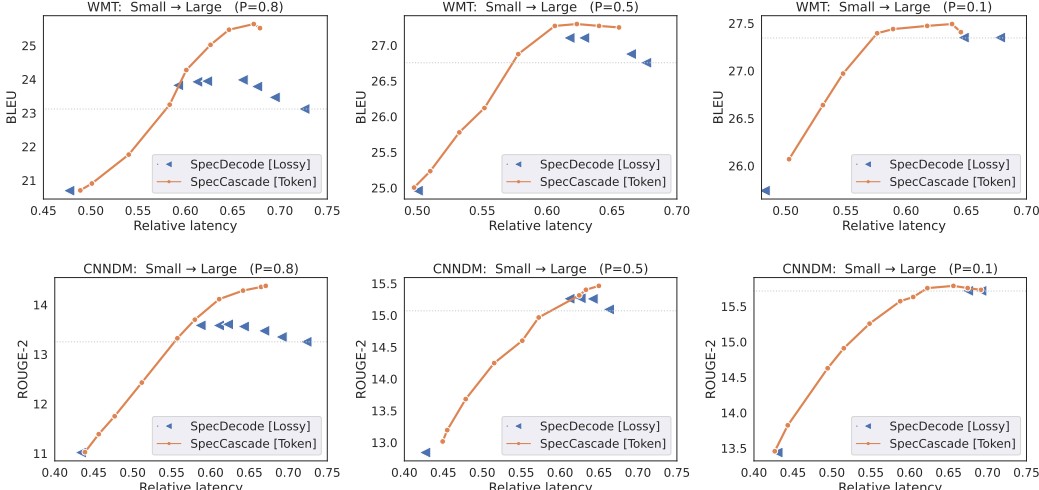

Figure 7: Plots of quality vs. latency for T5 models under **Top-$P$ sampling with varying values of** $P$. Each method interleaves T5-small with T5-large. The $x$-axis tracks the latency *relative* to that of calling the large model on all inputs. The horizontal dotted line denotes the quality of the large model. As $P$ becomes smaller, SpecDecode [Lossy] is able to produce fewer unique trade-off points as the scaling by the trade-off parameter $\alpha$ is less effective in its acceptance criterion (§C.3). SpecCascade [Token], which uses the TokenV3 deferral rule in (15), does not suffer from the same issue; here the trade-off parameter does *not* feature as a scaling term in the acceptance criterion.

and pick the value of the threshold $\alpha$ on this metric from the range $[0, 10]$. For temperature sampling with a non-zero temperature, we use the following natural analogue to the above $D$:

$$D(q, p) = -\mathbb{E}_{v \sim q}\left[\log(p(v))\right] = -\sum_{v \in \mathcal{V}} q(v) \cdot \log(p(v)).$$

In §F.5, we present comparisons between different implementations of this method.

**Lossy speculative decoding.** See §F.6 for details.

## F.2 COMPARISONS UNDER VARYING TEMPERATURES AND GREEDY DECODING

In Figures 5 and 6, we provide additional plots of quality vs. latency for different inference strategies under greedy decoding ($T = 0$) and temperature sampling ($T = 0.1, 0.5, 1.0$) respectively.

As expected, we see from the greedy decoding results in Figure 5 that all methods yield better quality metrics compared to their performance under temperature sampling. In this case, the proposed SpecCascade [TokenV3] is equivalent to the lossy variant of speculative decoding proposed by Leviathan et al. (2023) for greedy decoding, which we refer to as SpecDecode [Lossy, Greedy]. See §C.4 for the discussion of this equivalence. Similarly, as noted in §D.4, with greedy decoding, the OPT deferral rule coincides with the Diff deferral rule.

In the temperature sampling results in Figure 6, we compare the lossy speculative sampling proposed by Leviathan et al. (2023) for temperature sampling (SpecDecode [Lossy]) with two of our proposed methods: speculative cascades with the OPT and TokenV3 (referred to as Token) deferral rules. We can see that SpecCascade [Lossy] generates fewer unique trade-off points as temperature $T$ gets smaller. As noted in §C.1, this is because its acceptance criterion is less susceptible to changes in trade-off parameter $\alpha$ when $\mathbb{S}(p_t)$ is peaked. In contrast, our proposed SpecCascade [Token] approach yields a wider range of trade-off points; here the trade-off parameter is not used to construct a traded-off target distribution, and does not feature as a scaling term in the acceptance criterion.

The reason the Token-specific rule fares better than OPT is because the latter computes its deferral decisions based on which of $q_t(\cdot)$ and $p_t(\cdot)$ is more *peaked*; this can be a disadvantage when the

Table 5: Reduction in latency ($T = 1, \gamma = 5$) when matching the quality of the large model, and the best quality metric without exceeding the latency of the large model. We use T5-small and T5-large as the small and large models respectively. Quality is measured in terms of the BLEU for WMT and ROUGE-2 for CNNDM. We apply Top-$P$ sampling with varying values of $P$. The proposed `SpecCascade [Token]` method uses the `TokenV3` deferral rule in (15).

| Method | Latency↓ when matching large model's quality | | | | | | Best quality *without* exceeding large model's latency | | | | | |
|---|---|---|---|---|---|---|---|---|---|---|---|---|
| | $P = 0.1$ | | $P = 0.5$ | | $P = 0.8$ | | $P = 0.1$ | | $P = 0.5$ | | $P = 0.8$ | |
| | WMT | CNNDM | WMT | CNNDM | WMT | CNNDM | WMT | CNNDM | WMT | CNNDM | WMT | CNNDM |
| SpecDecode [Lossy] | 1.55× | 1.48× | 1.64× | 1.63× | 1.69× | 1.75× | 27.35 | 15.72 | 27.10 | 15.27 | 23.98 | 13.61 |
| SpecCascade [Token] | **1.74×** | **1.61×** | **1.73×** | **1.65×** | **1.71×** | **1.79×** | **27.49** | **15.79** | **27.30** | **15.46** | **25.64** | **14.38** |

sampled token is not be close the distribution mode, which is likely to happen with higher temperatures. With lower temperatures, however, the sampled token is likely to be close the distribution mode, and as a result, the advantage that the `Token`-specific rule has over `OPT` diminishes.

## F.3 COMPARISONS UNDER TOP-$P$ SAMPLING

We compare speculative cascades with the `TokenV3` deferral rule (`SpecCascade [Token]`) with `SpecDecode [Lossy]` under top-$P$ sampling. As discussed in §C.3, when implementing `SpecCascade [Token]`, the `TokenV3` deferral rule in (15) is applied to the original drafter and verifier distributions; we use the deferral decision to then interleave top-$P$ truncated versions of these distributions. In Table 5, we report the results of our evaluation for varying $P$, while fixing the temperature $T = 1$ and $\gamma = 5$.

It is for the smallest value of $P$ that our proposal offers the largest gains in speed-up over `SpecDecode [Lossy]`. Unsurprisingly, the smaller the value of $P$, the better are the quality metrics, with both methods being almost quality neutral for $P = 0.1$.

As we elaborate in §C.3, as $P$ becomes smaller, the trade-off parameter $\alpha$ in lossy speculative sampling becomes less effective, with the method becoming identical to standard *loss-less* speculative decoding when $P \to 0$. In contrast, our proposed `SpecCascade [Token]` approach does not suffer from the same issue as it uses the trade-off parameter $\alpha$ not as a scaling parameter in the acceptance criterion, but to construct a new target distribution that is amenable to a higher acceptance rate even under top-$P$ sampling. Therefore we are able to tune $\alpha$ to get a wider range of operating points and match the quality of the larger model at a lower latency.

For example, in Figure 7, lossy speculative decoding with $P = 0.1$ is able to offer only three unique trade-off points (despite sweeping through a fine-grained grid on $\alpha$ from $10^{-6}$ to 1); in contrast `SpecCascade [Token]` is able to offer a wider range of trade-off points.

## F.4 COMPARISONS UNDER DIFFERENT BLOCK SIZES $\gamma$

In Figure 8, we present latency-quality trade-off plots for T5 cascades under different block sizes $\gamma$. In each case, we find that the proposed speculative cascading techniques outperform lossy speculative decoding across different latency values. Furthermore, higher values of $\gamma$ are seen to yield a wider range of trade-offs, with lower quality operating points shifting to the left, and better quality operating points shifting to the right. For example, with XSum, `SpecDecode [Lossy]` with $\gamma = 3$ matches the small model's quality at 0.64 relative latency, and matches the large model's quality at 0.85 relative latency; with $\gamma = 7$, it matches the small model's quality at an even lower latency, but practically provides no speed-up when matching the larger model's quality. The reason a larger block size can hurt speed-up at the higher quality regime is because it can result in frequent rollbacks, thus defeating the purpose of using speculative execution.

## F.5 BIG LITTLE DECODER (BILD) VARIANTS

In our experiments in the main text (§6), we compared against a version of the Big Little Decoder method (Kim et al., 2023) that applied Algorithm 4 to the target distribution $\mathbb{T}_{\text{BiLD}}$ the authors seek to mimic (§B). We now show that this version performs similarly to the original BiLD algorithm in (Kim et al., 2023).

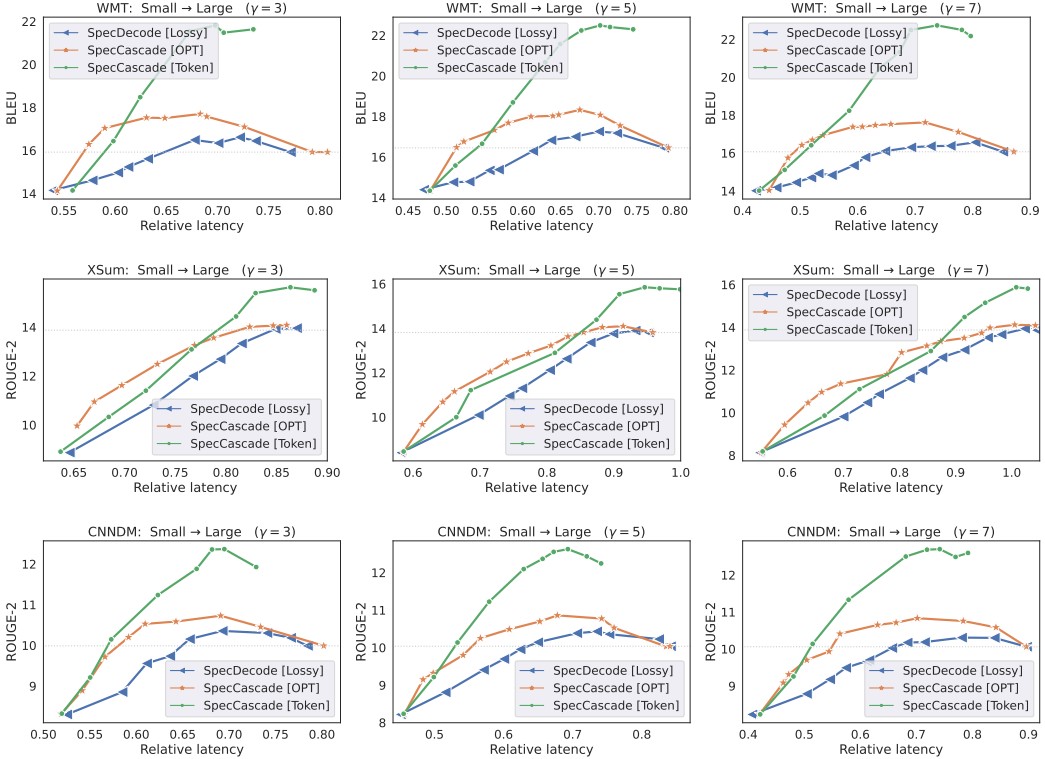

Figure 8: Plots of quality vs. latency for T5 models with **with varying block sizes** $\gamma$. Each method interleaves T5-small with T5-large. The $x$-axis tracks the latency *relative* to that of calling the large model on all inputs. The horizontal dotted line denotes the quality of the large model.

A key difference to the original algorithm in (Kim et al., 2023) is the use of the fallback phase, where the drafter is run until its maximum predicted probability $\max_{v \in \mathcal{V}} q(v) < 1 - \alpha_f$, for a threshold $\alpha_f \in [0, 1]$ (or until a maximum block size of 10 is reached), and the use of a deterministic rollback policy where the verifier rejects a draft token whenever $D(q, p) > \alpha$. In our implementation, we adopt the speculative sampling algorithm from (Leviathan et al., 2023): we do not have the fallback policy and replace the determinisic rollback policy with the rejection sampling in Algorithm 4.

Figure 9 (top) compares the original version of BiLD with the version we use in §6. We interleave between a T5-small and T5-large model on WMT, using greedy decoding ($T = 0$) for inference. As prescribed by the authors (Kim et al., 2023), we use the following discrepancy metric for greedy decoding:

$$D(q, p) = \log p \left( \arg\max_{v \in \mathcal{V}} q(v) \right).$$

We compare our implementation (BiLD*), where we set the block size to 5 (same as our proposed speculative cascading approaches), with the original BiLD for different choices of maximum block size $\gamma$ and different fallback thresholds $\alpha_f$. For both methods, we vary the threshold $\alpha$ on $D(q, p)$ to vary the latency and plot the resulting BLEU score.

A higher fallback threshold $\alpha_f$ results in larger draft generation windows; this gives an advantage in the low latency regime, where most of the draft tokens are accepted. As a result, BiLD [$\gamma = 10, \alpha = 0.9$] yields the lowest latencies, but also yields lower quality. A low fallback threshold results in very small draft generation windows, and consequently, in higher latencies. This is why BiLD [$\gamma = 5, \alpha = 0.1$] is the slowest but yields high quality metrics.

Our implementation BiLD* is seen to perform comparable to the best parameter choices for the original BiLD algorithm in Figure 9.

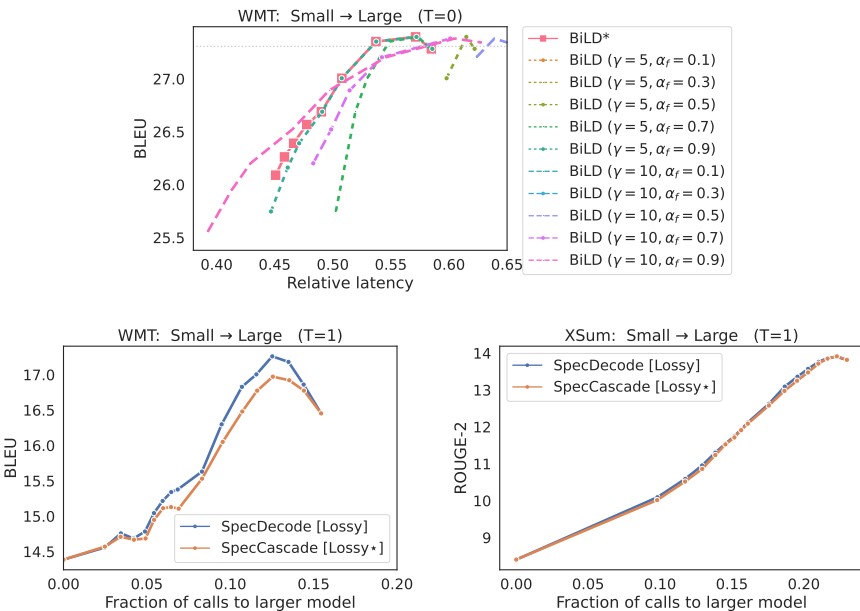

Figure 9: Top: Plots of quality vs. latency **comparing BiLD\* with the original BiLD algorithm in** Kim et al. (2023) with varying maximum draft window size $\gamma$ and fallback confidence threshold $\alpha_f$. Bottom: Comparison of lossy speculative decoding with $\beta = 1$ [Lossy] and $\beta$ tuned using the procedure in (Tran-Thien, 2023) [Lossy\*].

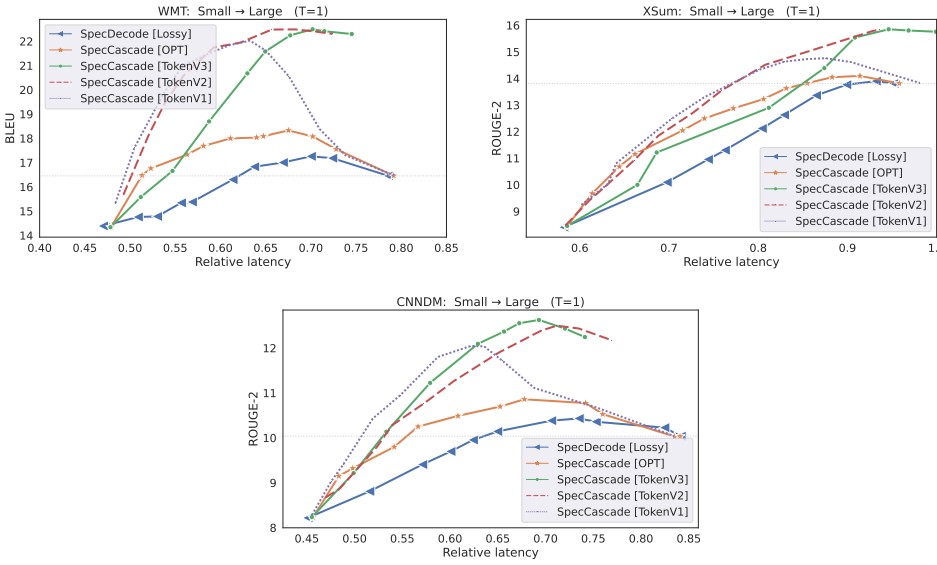

Figure 10: Plots of quality vs. latency for **T5 models with all three token-specific speculative cascade deferral rules** in equations 13–15. Each method interleaves a T5 small and a T5 large model. The $x$-axis tracks the latency *relative* to that of calling the large model on all inputs. The horizontal dotted line denotes the quality of the large model.

It is worth noting that while we view $\mathbb{T}_{\text{BiLD}}$ as the target distribution that the algorithm in (Kim et al., 2023) seeks to mimic, the presence of the fallback phase could mean that on some inputs a output response is generated without the verification (or rollback) phase being invoked. In such cases, the output response will come solely from drafter if it turns out that it contains tokens for which $D(q_t, p_t) > \alpha$.

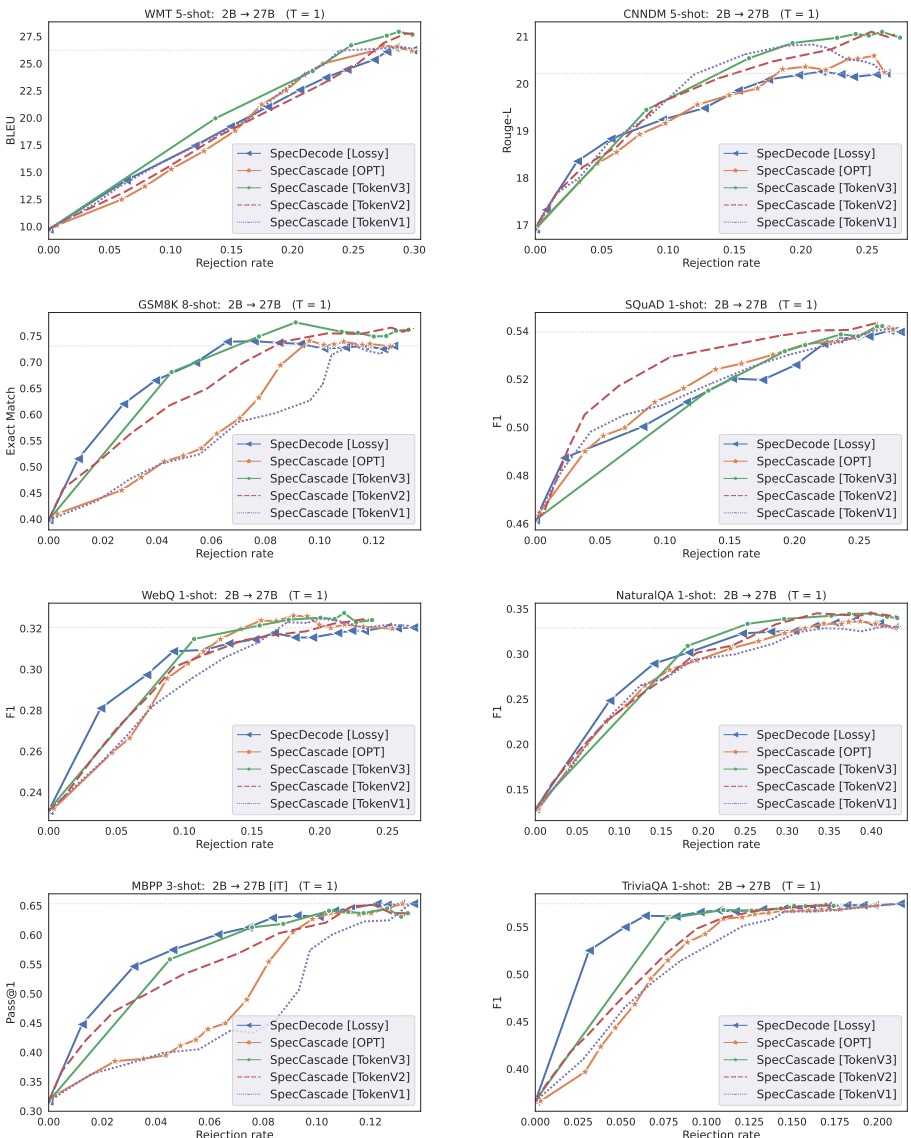

Figure 11: Plots of quality vs. rejection rate for **Gemma models with all three token-specific speculative cascade deferral rules** in equations 13–15. Each method interleaves a Gemma 2B drafter with a Gemma 27B verifier. The horizontal dotted line denotes the quality of the large model. We include all three token-specific speculative cascade deferral rules in equations 13–15.

## F.6 LOSSY SPECULATIVE DECODING VARIANTS

In our experiments in the main text (§6), we compared against the lossy speculative decoding (Tran-Thien, 2023; Zhou et al., 2024) described in §2, with the parameter $\beta$ set to 1. We now present results for this method with $\beta$ tuned according to the procedure in Tran-Thien (2023), and show that choosing $\beta = 1$ fares at least as well as tuning $\beta$.

The goal in Tran-Thien (2023) is to choose $\alpha$ and $\beta$ so as to maximize the acceptance rate for the draft token, while ensuring that the KL divergence between the resulting target distribution and $p$ is within an allowable limit $R$. The authors prescribe specifying $R$, and for each prefix, tuning $\alpha$ and $\beta$ to solve the resulting constrained optimization problem. To be consistent with the rest of our experimental setup, we vary $\alpha$ to vary the draft acceptance rate (note that each choice of $\alpha$ corresponds to a particular KL divergence to $p$), and tune $\beta \geq 1 - \alpha$ to satisfy the following condition

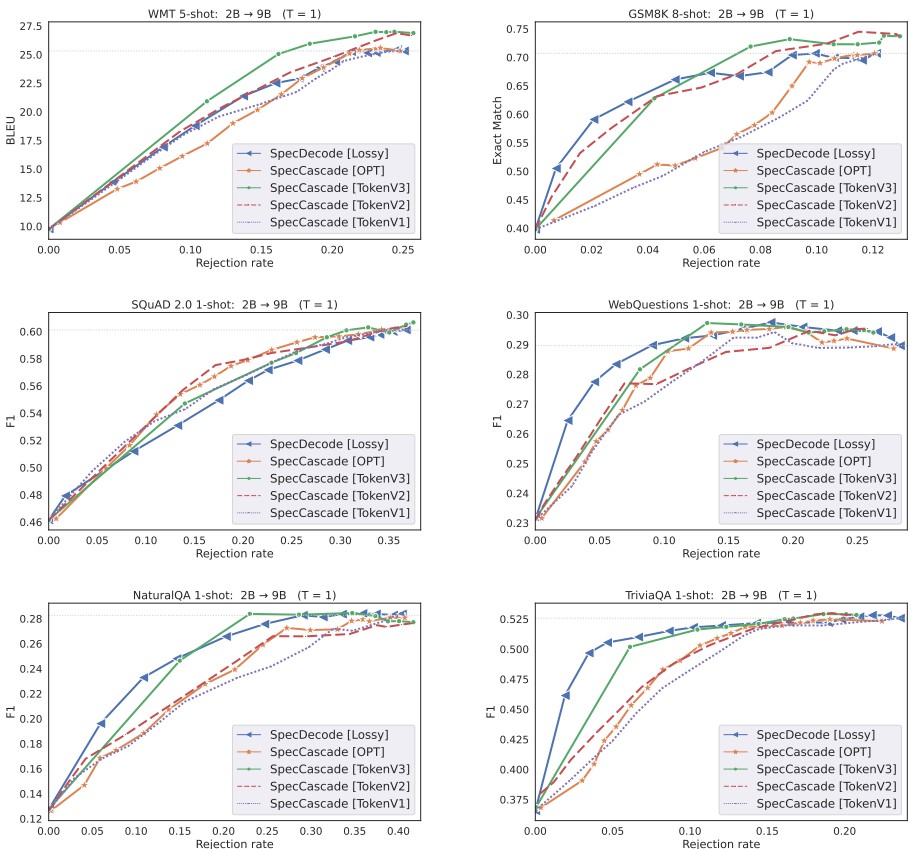

Figure 12: Plots of quality vs. rejection rate with **Gemma 2B → 9B** speculative cascades. Each method interleaves a Gemma 2B drafter with a Gemma 9B verifier. The horizontal dotted line denotes the quality of the large model. We include **all three token-specific speculative cascade deferral rules in equations 13–15**.

outlined in Tran-Thien (2023):

$$\sum_{v \in \mathcal{V}} \max \left\{ 0, q(v) - \frac{p(v)}{1-\alpha} \right\} = \sum_{v \in \mathcal{V}} \max \left\{ 0, \frac{p(v)}{\beta} - q(v) \right\}$$

We pick $\beta$ using a grid-search over 1000 values between $\alpha$ and 10.

Since this tuning procedure, in turn, can add to the method's latency, for a fair comparison, we plot quality as a function of the fraction of calls to the large model (rejection rate), instead of relative latency. In Figure 9 (bottom), we plot these trade-off curves for loss speculative decoding with $\beta = 1$ (Lossy) and for speculative decoding with $\beta$ tuned using the above procedure (Lossy$^\star$). We compare performances on WMT and XSum, and in each case, interleave a T5-small model with a T5-large model.

In both cases, setting $\beta = 1$ provides trade-offs comparable to or better than using a tuned value of $\beta$. The reason using a tuned value of $\beta$ fares worse than setting $\beta = 1$ might be because we are measuring quality in terms of BLEU or ROUGE-2, which is different from the KL divergence to $p$ objective that the tuning procedure in Tran-Thien (2023) seeks to optimize.

### F.7 TOKEN-SPECIFIC DEFERRAL RULE VARIANTS

In Figure 10, we present latency-quality trade-off plots for cascades constructed from a T5-small and a T5-large model. We include in these comparisons, all three token-specific deferral rules in

(13)–(15). In Figure 11, we present trade-off plots for cascades constructed from Gemma 2B and Gemma 27B models with all three token-specific rules, and in Figure 12, we include similar plots for cascades constructed from Gemma 2B and Gemma 9B models. We note that the trends with the 2B $\to$ 9B are similar to those seen with the 2B $\to$ 27B cascades.

With the T5 models, the results are mixed, with the V1 and V2 variants sometime surpassing the V3 variant (which is the variant we included in the main experiments results in §6). Interestingly, with the Gemma models, the V3 variant is seen to outperform the others for most rejection rates, with the exception of the 2B→27B cascade on SQuAD 2.0, where the V2 variant is better.

The reason for the V3 variant outperforming the V1 and V2 variants on the Gemma models could be due to the fact that it uses the larger model's distribution $p_t(\cdot)$ to measure confidence for both the drafter and verifier tokens (see LHS and RHS in (13)). We expect this to be particularly helpful when there is a larger gap in sizes between $q$ and $p$, and the larger model's distribution is better aligned with the data-generating distribution compared to the smaller model. Furthermore, as per the discussion in §5, the multiplicative form of the rule (15) results in a target distribution with an intuitive form: it seeks to mimic $q_t(\cdot)$ on the top-$\alpha$ ranked tokens by $p_t(\cdot)$ and uses a re-scaled version of $p_t(\cdot)$ for the other tokens:

$$\pi_{\texttt{TokenV3}}(v) = q_t(v) \cdot \mathbf{1}\big(v \in \mathcal{T}_\alpha\big) \, + \, p_t(v) \cdot \sum_{v' \notin \mathcal{T}_\alpha} q_t(v'),$$

where $\mathcal{T}_\alpha = \{v \in \mathcal{V} : p_t(v) \geq \max_{v'} p_t(v') \cdot (1 - \alpha)\}$.

## G    LIMITATIONS

One of the limitations of our proposal is the use of plug-in estimators to approximate the optimal rule (9). While these approximations are effective in practice, they rely on the individual models being calibrated. An alternative to the use of plug-in estimators is to use a router model explicitly trained to mimic the optimal rule using a validation sample drawn from $\mathbb{P}$ (Gupta et al., 2024). Another limitation is that the optimization objectives we seek to minimize are local objectives that seek to make the best deferral decision at the current position $t$. In doing so, they ignore the downstream effects of choosing a particular model in the current step. Devising a global deferral objective that takes downstream errors into account would be an interesting direction for future work. More broadly, our paper seeks to improve cost-quality trade-offs in LM inference. It is important that such improvements do not unfairly advantage one slice of the data or a subset of the population, at the cost of others. Ensuring that the trade-off gains that our approach offers is equitable across different slices of the data is another important direction for the future.

