# OpenReview forum: "Faster Cascades via Speculative Decoding"
_ICLR.cc/2025/Conference — ICLR 2025 Oral_

### Official Review · Reviewer_Wu5Z · 2024-10-21

**Soundness:** 4
**Presentation:** 3
**Contribution:** 3
**Rating:** 8
**Confidence:** 3

**Summary:**

This paper studies way to interleave two adaptive computation methods: cascading and speculative decoding. The authors present a framework that combines both methods, and aims to enjoy the benefits of both approaches. They also derive theoretical guarantees as to the optimal deferral rule for their method. The key idea is replacing the target distribution in speculative decoding with a different distribution, which takes both the small and large distributions into account. Experimental results on various benchmarks show that the best two variants lead better speed-accuracy tradeoffs compared to either approaches.

**Strengths:**

- An interesting and intuitive approach: in cascading, the small model can sometimes outperform the larger one. In contrast, in speculative decoding, the model is guaranteed to match the large model quality, but is typically faster. By combining them, the authors allow for fast decoding, with potential improvement, which leads to overall higher speedup.
- Some of the claims are clever. I particularly like the intuition that only considering the confidence of the small model is sub-optimal, and we should also take the large model's confidence into account, and how this could be implemented in speculative decoding (Remark 2).
- The best proposed method (SpecCascade [Token]) seems to outperform all baselines quite consistently.

**Weaknesses:**

* I had some trouble following section 4.3. A roadmap/intuition would have been helpful. In particular, I did not fully understand the role of Lemma 3, and the overall takeaway from this section.

- The experiments section was also a bit hard to follow. It starts with outlying the different deferral rules, and then presents the baselines. It seems both parts are somewhat overlapping. It would be helpful to merge them and discuss the link between the two, and particularly not have a paragraph separating the two.

- It is not entirely clear from table 2 which lines are the current work and which are previous work. I think only the last two lines are new, but the line separating the different groups appears before the last four lines. Also, the name SpecDecode (which hints of the current work) also appears earlier (SpecDecode [Lossy]). I think this name stems from the new perspective presented in this work, but it is still quite confusing and makes evaluating the results challenging. I would suggest clearly separating existing and new methods both visually (within the table) and by name to avoid confusion.

**Questions:**

- The intuition behind 4.4 (the first paragraph in that section) is not entirely clear to me. Can you please explain in more detail the problem you are trying to address here?

---

> ### Author Response · Authors · 2024-11-20
> **Response to Reviewer Wu5Z**
>
> We thank the reviewer for the positive comments. Below, we provide detailed clarifications for the questions raised.
>
> ___
> > Roadmap / intuition for Section 4.3
> ___
>
> Thanks for the suggestion. We have updated the manuscript to include the following high-level overview.
>
> The main goal in this section is to construct a deferral rule $r(\cdot)$ for a speculative cascade that takes a prefix as input and decides whether the small or the large model is better suited to predict the next token.
>
> We propose choosing the deferral rule by minimizing two competing objectives:
> - (a)  the expected loss against the ground-truth distribution (quality)
> - (b)  the rejection rate for the resulting target distribution under speculative execution (inference cost)
>
> The section follows the roadmap below:
> - In Lemma 3, we show that the the rejection rate in (b) can be computed using a simple closed-form expression.
> - In Equation 7, we devise a constrained optimization problem that combines (a) and (b) by minimizing (a) with a constraint on (b); the constraint uses the expression in Lemma 3.
> - In Equation 8, we formulate the Lagrangian for the constrained optimization problem.
> - In Lemma 4, we derive the deferral rule that optimizes the Lagrangian.
> - In Equation 10, we approximate the optimal deferral rule with a plug-in rule that is feasible to compute in practice.
> - In Lemma 5, we provide a theoretical guarantee for the plug-in approximation.
>
> ___
> > Merge discussion on deferral rules and baselines in experiments section
> ___
> In the uploaded revised manuscript, we have merged the two subsections.
>
> ___
> > Table 2: demarcation between proposed methods and baselines
> ___
> Rows 1–4 are the baselines. Rows 5–8 are the new speculative cascading methods proposed in this paper.
>
> More specifically, the baselines we compare to are:
> - SeqCascade [Chow]
> - TokenCascade [Chow]
> - SpecDecode [Lossy]
> - BiLD*
>
> Of these, the first baseline operates at the sequence level and the rest operate at the token level.
>
> The proposed methods include:
> - SpecCascade [Chow]
> - SpecCascade [Diff]
> - SpecCascade [OPT]
> - SpecCascade [Token]
>
> Of these, the first two are based on deferral rules that are already used with sequential cascades (sections 3.1 and 3.2). The last two are based on new deferral rules that are designed in section 4.2 and 4.3.
>
> In the uploaded revised manuscript, we have clearly separated these groups, and clarified the differences in the table caption.
>
> ___
> > SpecDecode [Token] in Figure 3
> ___
>
> Apologies for the confusion. This is a typo and should be **SpecCascade [Token]**, which is a new method proposed in this paper (specifically, a speculative cascade with the deferral rule in Equation 13). We have corrected this in the revised manuscript.
>
> ___
> >  Intuition behind Section 4.4 and the first paragraph
> ___
>
> Up until this section, the deferral rules presented in the paper work as follows: they take a prefix as input, and choose between the small model’s distribution $q_t$ and the large model’s distribution $p_t$ to sample the next token. In Section 4.4, we explore **new deferral rules that interleave between $q_t$ and $p_t$ in a smarter manner** (specifically, constructs a new distribution that combines $q_t(\cdot)$ and $p_t(\cdot)$ at the token level).
>
> The first paragraph points out that deferral rules so far make their decision by comparing the maximum token probabilities from the two distributions. A downside to doing this is that despite picking the distribution with the higher max probability, we may still end up sampling a token from a low-probability region of the distribution. To avoid this drawback, we construct a new interleaved distribution which mimics $q_t(v)$ on a subset of tokens deemed to be of acceptable quality and is a re-scaled version of $p_t(\cdot)$ otherwise.

---

> > ### Comment · Reviewer_Wu5Z · 2024-11-21
> >
> > Thank you for your response, my concerns have been addressed.

---

> > > ### Author Response · Authors · 2024-12-02
> > > **Thank you!**
> > >
> > > Dear Reviewer,
> > >
> > > We are glad that your concerns have been addressed. Thanks a lot for the positive feedback!

---

### Official Review · Reviewer_ySgY · 2024-10-30

**Soundness:** 3
**Presentation:** 4
**Contribution:** 3
**Rating:** 6
**Confidence:** 2

**Summary:**

This work investigates the integration of two front-ended inference strategies: model cascades and speculative decoding. It introduces a novel decoding strategy called speculative cascading. This method combines the strengths of both strategies to achieve a more favorable cost-quality trade-off during inference. Experiments with Gemma and T5 models across  various benchmarks (e.g., summarization, translation, coding, and reasoning) demonstrate the effectiveness of this method, which yields better cost-quality trade-offs than cascading and speculative decoding baselines.

**Strengths:**

1. The paper addresses a promising and challenging research direction by combining model cascades with speculative decoding.  Recent studies have shown interest in integrating speculative decoding with advanced techniques, such as contrastive decoding [1], to accelerate inference while enhancing the generation quality of LLMs. Speculative cascading complements these efforts by exploring model cascades in speculative decoding. Through both empirical and theoretical analyses, this work innovatively integrates various deferral rules within a speculative execution framework, which is non-trivial and could provide valuable insights for the the academic community.
2. The design of speculative cascading is thorough and well-motivated. The authors provide a clear exposition of the theoretical foundations for both model cascades and speculative decoding, which they summarize effectively in Table 1. Speculative cascading is systematically crafted to leverage the strengths of each of these techniques.
3. The authors conduct extensive experiments with the Gemma and T5 models, carefully detailing experimental settings and effectively validating the efficacy of speculative cascading. The results demonstrate that speculative cascading achieves better cost-quality trade-offs compared to conventional cascades and speculative decoding methods.
4. The manuscript is clearly written, with a well-structured narrative, compelling motivation, detailed analyses, and transparent demonstrations that enhance its readability and impact.

**Weaknesses:**

1. **Fairness of Comparison**: In Table 2, the authors report minimal latency when matching the quality of a large model, as well as the best quality metric achievable without exceeding the latency of LLMs for each method. However, it is unclear if these comparisons are entirely fair. For instance, it would be helpful to know if the results for BiLD were reported under similar configurations, ensuring a consistent basis for comparison.
2. **Applicability of Speculative Cascading**: Figures 2 and 3 suggest that the quality of speculative cascading can be significantly affected by relative latency. The manuscript would benefit from a more detailed discussion on optimizing the cost-quality trade-off during inference. Specifically, guidance on configurations for maximizing speed-ups while maintaining quality or improving quality without exceeding the latency of the original LLMs, would enhance understanding and practical applicability.
3. **Quality Improvement of Speculative Cascading**: As shown in Figures 2 and 3, the quality improvements of speculative cascading are relatively modest across several tasks, including WMT 5-shot, GSM8K 8-shot, WebQ 1-shot, and NaturalQA 1-shot with Gemma models. This limited improvement in quality may constrain the broader applicability of speculative cascading. Additional discussion on scenarios where speculative cascading performs optimally would provide valuable context for readers.

**Questions:**

Most of my primary concerns are outlined in the weaknesses section above. Here is an additional minor concern:

- In line 527, the authors state, “a lower rejection rate directly translates to a lower latency.” It would be helpful to provide a direct demonstration of this relationship, showing the correlation between latency and rejection rate for clarity.
- As I am not a specialist in the theoretical domain, I cannot fully assess the accuracy of the theoretical analysis presented in this manuscript. My opinion on this aspect may change after reading insights from other reviewers or engaging in further discussions.

---

> ### Author Response · Authors · 2024-11-20
> **Response to Reviewer ySgY**
>
> We thank the reviewer for the encouraging comments. Below, we address the questions raised.
> ___
> > guidance on configurations for maximizing speed-ups while maintaining quality or improving quality without exceeding the latency of the original LLMs.
> ___
>
> This is a great question! The proposed method has a **single threshold parameter $\alpha$** that can be tuned to achieve a desired latency-quality trade-off. In applications where we are provided a maximum allowable latency, we can pick a threshold that achieves the best quality for the specified latency. This is typically the case with applications of cascades, where it may be acceptable to slightly sacrifice quality for latency gains. Alternatively, a practitioner may want to pick the threshold to match the quality of the large model at the least possible latency.
>
> In some applications, one may want to allow the customer to pick a suitable trade-off at the time of deployment; in this case, we could maintain a lookup table of multiple operating points and associated thresholds, and have the customer pick the one most suitable to them at inference time.
>
> ___
> >  it would be helpful to know if the results for BiLD (in Table 2) were reported under similar configurations, ensuring a consistent basis for comparison
> ___
>
> All methods in Table 2 use the same block size and temperature, and have a single **cost/threshold parameter $\alpha$ that we sweep to achieve different latency-quality trade-offs** (see Table 1). For each method, we report the best quality achieved for the same latency as the large model and the best latency achieved for the same quality as the large model.
>
> The BiLD* method we report in Table 2 also has a single trade-off parameter to tune. As detailed in Section 5, this implementation of BiLD is slightly different from the original algorithm proposed by Kim et al. (2023), in that it uses a fixed draft window of size $\gamma$. In contrast, the original algorithm uses a **dynamic draft window**, with the drafter run until its maximum predicted probability falls below a *fallback* threshold.
> For a fair comparison, all methods in the main text use a fixed-sized draft window. In Appendix E.5, we report additional comparisons to the original BiLD algorithm using a dynamic draft window; in this case, we tune two threshold parameters, and compare the resulting trade-offs in Figure 8.
>
> ___
> >  Improvements with Gemma models relatively modest and scenarios where speculative cascading performs optimally
> ___
>
> While some of the gains on Gemma models are less pronounced than T5, we note that on challenging tasks such as MBPP, we still see notable gains over speculative decoding.
> More generally, we expect speculative cascades to yield significant gains over speculative decoding when there exists a slice of data where the small model performs comparable to or better than the large model. The larger this slice, the larger is the improvement over speculative decoding. We will be happy to include a discussion on this in the paper.

---

> ### Comment · Reviewer_ySgY · 2024-11-26
>
> I appreciate the authors' detailed responses addressing my previous concerns. The clarifications and additional explanations have resolved the points I raised. I suggest incorporating these helpful discussions into the revised manuscript.
>
> Overall, I maintain **a positive recommendation** for this work based on its empirical novelty and well-designed experimental methodology. However, as my expertise lies primarily in empirical aspects rather than theoretical foundations, **I recommend that the area chair carefully evaluate the theoretical analysis and related discussions between the authors and other reviewers**.
>
> Thank you for your hard work. Good luck!

---

> > ### Author Response · Authors · 2024-12-02
> > **Thank you!**
> >
> > Dear Reviewer,
> >
> > We are happy that we were able to clarify and resolve the points you raised. Thanks a lot for the positive recommendation and the appreciation of the paper's "empirical novelty and well-designed experimental methodology".

---

### Official Review · Reviewer_jr8P · 2024-11-02

**Soundness:** 1
**Presentation:** 3
**Contribution:** 1
**Rating:** 3
**Confidence:** 4

**Summary:**

The paper introduced new speculative decoding variations by combining two techniques: speculative decoding and cascade language models. This is achieved by applying cascade rules on the token level. Experiments showed that the proposed method achieves faster decoding and performs better when the temperature is high.

**Strengths:**

- The proposed method is lightweight and does not require supervised fine-tuning.
- The paper flows naturally and is easy to read and understand overall.

**Weaknesses:**

- The reasoning behind the designs of the loss functions in Equations (3) and (8) is unclear.
- The experimental design seems unfair and the improvements are limited. SpecCascade [Token] uses top tokens, while the other methods are evaluated with vanilla sampling at a temperature of 1. Sampling methods like Top-K and Top-P can lead to better performance by avoiding out-of-distribution tokens. To ensure a fair comparison, baseline methods should also use Top-K or Top-P sampling. Given the experiment in Figure 5, which shows that SpecCascade does not improve over lossy SD when T=0, I doubt whether the proposed method offers meaningful enhancements other than inserting a logic removing out-of-distribution tokens.
- The evaluations in Table 2 and Figure 2 use gamma=5, which is not optimal; speculative decoding typically requires hyperparameter tuning for the best results.
- The sub-figures in Figure 3 (and other figures with sub-figures) are not properly aligned.

**Questions:**

1. What is the ground truth (mentioned in lines 76, 156, 166, etc) in this paper? Why would you optimize toward the ground truth probabilities?
2. Line 353, what is OPT?
3. What hardware is used for the experiment? How are the models implemented?
4. What is SpecDecode [Token] in Figure 3?

---

> ### Author Response · Authors · 2024-11-20
> **Response to Reviewer jr8P (Part 1)**
>
> We thank the reviewer for their comments and questions. We have provided elaborate clarifications below and are happy to address any further questions.
>
> ___
> > SpecCascade [Token] uses top tokens, while the other methods are evaluated with vanilla sampling at a temperature of 1.
> ___
> We believe there is a misunderstanding here: ***all the methods*** compared (including SpecCascade [Token]) ***use sampling with a fixed temperature***.
>
> **Choice of target distribution vs. sampling technique:**
>
> The key distinction of our proposal to speculative decoding is *how the target distribution is chosen*; it is **not how tokens are drawn** from the target distribution.
>
> Specifically, while ***speculative decoding samples from the large model’s distribution $p_t$, our proposal is to sample from a target distribution that interleaves the small model’s distribution $q_t$ with $p_t$***. Indeed our main contribution is the design of different interleavings between $q_t$ and $p_t$ that offer better cost-quality trade-offs compared to standard speculative decoding.
>
> **Target distribution for SpecCascade [Token]:**
>
> What the reviewer may be referring to is the specific interleaving between $q_t$ and $p_t$ employed in SpecCascade [Token], which assigns $q_t(v)$ to tokens $v$ that are ranked high by $p_t$ and assigns a re-scaled version of $p_t$ otherwise (line 391 in paper):
>
> $$\pi(v) = q_t(v) \cdot  \mathbf{1}\big(v \in A\big)  + p_t(v) \cdot \sum_{v' \notin A} q_t(v')$$
> where $A$ is the set of top ranked tokens by $p_t$, i.e., $A = \\{v \in \mathcal{V}: p_t(v)  \geq \max_{v'} p_t(v') \cdot(1 - \alpha)\\}.$
>
> Intuitively, the target distribution mimics $q_t(v)$ on tokens $v$ that are ranked high by $p_t$; if not, we simply revert to mimicking $p_t(\cdot)$. In the extreme case where $q_t$ assigns no mass to the top ranked tokens by $p_t$, the target distribution defaults to sampling from $p_t$, i.e. $\pi(v) = p_t(v)$. The rationale behind this choice is to *ensure that the target distribution does not deviate much from $q_t$ (so that the draft rejection rate is low), while also ensuring that the sampled token is of acceptable quality*.
>
> **Differences to top-p sampling:**
>
> **Our choice of $\pi$ is *not* the same as applying top-p sampling to the large model’s distribution, which would sample only tokens that are top-ranked by $p_t$**. In particular, top-p sampling samples token $v$ with probability:
>
> $\frac{1}{Z} \cdot  p_t(v) \cdot \mathbf{1}\big(v \in A\big),$
>
> where $Z = \sum_{v’} p_t(v’) \cdot  \mathbf{1}\big(v’ \in A\big)$.
>
> Notice that top-p sampling amounts to assigning zero probability to tokens that are *not* top-ranked by $p_t$. In contrast, we interleave both $p_t$ and $q_t$ in a non-trivial manner,  and *do not discard a specific subset of tokens*. In particular, our approach is **not designed to discard out-of-distribution tokens**.
>
> ***We will include a graphical illustration in the paper explaining the differences between our target distributions and top-p sampling.***
>
> We additionally note that our paper also explores **other forms of interleaving** between $q_t$ and $p_t$, such as the OPT rule in Equation 12, or the token-specific interleavings in Equations 13 and 14 (Figure 9-11 in Appendix E.7).
> ___
> > What is the ground truth (mentioned in lines 76, 156, 166, etc) in this paper? Why would you optimize toward the ground truth probabilities?
> ___
>
> The ground-truth probability distribution $\mathbb{P}$ is simply **the data-generating distribution that the language model attempts to mimic**. Formally, it is an underlying distribution over the space of all possible sequences generated from the vocabulary. For example, this could be a distribution over prompt-response pairs in a question-answering or machine translation task, with sequences that are suitable responses to a prompt receiving higher probability than those that are not.
>
> In practice, we are provided samples from this ground-truth distribution, which we then use to either pre-train or fine-tune a language model (LM). An ideal LM would be one that accurately mimics $\mathbb{P}$, i.e. for any sequence of tokens $x_1 x_2 \ldots x_{t-1}$, accurately outputs the probability of the next token: $\mathbb{P}(\cdot | x_{<t})$.
>
> One typically measures the quality of an LM $p$ based on its agreement with $\mathbb{P}$. A common approach is to evaluate the log loss between $p$ and $\mathbb{P}$:
>
> $$
> \\mathbb{E} \left[ -\log(p(x_t | x_1, \ldots, x_{t-1}) \right].
> $$
>
> Given a sample $S$ of sequences drawn from $\mathbb{P}$, we may evaluate the empirical loss:
> $$
> \frac{1}{|S|} \sum_{(x_1, \ldots, x_t) \in S} -\log(p(x_t | x_1, \ldots, x_{t-1}).
> $$
>
> A **natural goal is to then learn an LM that minimizes the above loss against $\mathbb{P}$** (in the paper we also consider minimizing the expected 0-1 loss against the ground-truth distribution).
> ___

---

> > ### Author Response · Authors · 2024-11-20
> > **Response to Reviewer jr8P (Part 2)**
> >
> > ___
> > > The reasoning behind the designs of the loss functions in Equations (3) and (8) is unclear
> > ___
> > In this paper, we are provided two LMs $p$ (large) and $q$ (small), each of which mimics the ground-truth distribution $\mathbb{P}$ with varying accuracy.
> >
> > Speculative decoding seeks to simply re-produce the large model’s distribution $p$ without considering whether or not it is a good approximation to the ground-truth distribution $\mathbb{P}$. As a result, speculative decoding exactly matches the quality of the large model.
> >
> > In contrast, our paper seeks to **interleave $q$ and $p$ to construct a new target distribution that is a better approximation to $\mathbb{P}$** compared to both $q$ and $p$, i.e. is of better quality than both $p$ and $q$. We do so by explicitly minimizing a loss against $\mathbb{P}$.
> >
> > **Reasoning for Equation (3):**
> >
> > Equation (3) presents an objective that captures this goal. In this equation, we wish to learn a deferral function $r$ that for any prefix $x_{<t}$ either picks the small or large model’s output distribution over tokens, and we evaluate the loss of the resulting distribution against $\mathbb{P}$. More concretely, when we use the log loss to measure quality, the expected loss in equation (3) is given by:
> >
> > $$
> > \\mathbb\{E\} \left[ - (1 - r(x_{<t})) \cdot \log q(x_t | x_1, \ldots, x_{t-1}) + r(x_{<t}) \cdot -\log p(x_t | x_1, \ldots, x_{t-1}) \right],
> > $$
> > where when $r(x_{<t}) = 0$, the loss equals that for the small model, and when $r(x_{<t}) = 1$, the loss equals that for the large model.
> >
> > Equation (3) additionally introduces a cost parameter $\alpha$ denoting a cost for choosing the large model:
> > $$
> > \\mathbb\{E\}\_\{ \(x\_\{1\}, \\ldots, x\_\{t\}) \}  \left[ - (1 - r(x_{<t})) \cdot \log q(x_t | x_1, \ldots, x_{t-1}) + r(x_{<t}) \cdot (-\log p(x_t | x_1, \ldots, x_{t-1}) + \alpha)\right],
> > $$
> > We can then control the trade-off between inference cost and quality by varying the cost parameter.
> >
> > **Reasoning for Equation (8):**
> >
> > Equation (8) also crafts a similar objective, except that it additionally takes into account the fact that we will be sampling from the new target distribution through speculative execution. So instead of imposing a constant cost penalty for choosing $p$, we explicitly account for the rejection rate from using $q$ for drafting and the new target distribution for verification.
> > ___
> >
> > > The evaluations in Table 2 and Figure 2 use gamma=5, which is not optimal; speculative decoding typically requires hyperparameter tuning for the best results.
> > ___
> > **Appendix E.4** from the original manuscript includes a discussion on the **effect of block size $\gamma$.** Please see Figure 7 for a comparison of different block sizes. As with the experimental setup in Zhou et al. (ICLR 2024), we explore three different block sizes $\gamma = 3, 5, 7$, and find that in each case, the proposed techniques outperform lossy speculative decoding across different latency values.
> > ___
> > > The sub-figures in Figure 3 (and other figures with sub-figures) are not properly aligned
> > ___
> > Thanks for catching this. We have corrected this issue in the uploaded revised manuscript.
> > ___
> > > Line 353, what is OPT?
> > ___
> > OPT refers to the optimal deferral rule for the objective in Equation 8. In line 353, we present a plug-in approximation to the optimal deferral rule.
> > ___
> > > What hardware is used for the experiment?
> > ___
> > All methods are implemented on the same TPUv4 device. For the run-time analysis, we adopt the protocol followed in Leviathan et al. (2023); Zhou et al. (2024). We randomly sample 500 examples from the validation set, and calculate the wall-clock time taken for decoding.
> >  ___
> > > What is SpecDecode [Token] in Figure 3?
> > ___
> > Sorry for this typo. This should be **SpecCascade** [Token]. We have corrected this in the uploaded revised manuscript.

---

> ### Comment · Reviewer_jr8P · 2024-11-21
>
> Thank you for the clarification. However, I do not think my concerns are addressed. Specifically,
>
> 1. There is no confusion, and I do not mean your method is similar to top-p distribution. I suggested that your baselines should be stronger to prove the usefulness of your method. Users of LLMs will mostly use top-p or top-k distribution when performing sampling, which should be included as your baselines.
>
> 2.  With the recent advancement of LLMs, imitating a ground truth is not the main objective of language modeling, and reducing the loss is also not a proper objective of inference time optimization.
> Also, why sampling if the goal is to imitate the ground truth? Greedy decoding will result in higher performance. Combined with the fact that SpecCascade does not have a meaningful improvement in greedy decoding. I do not see the proposed method achieving the optimization goal of imitating ground truth.
>
> 3. Regarding the reasoning about equations 3 and 8, I do not think the theory is sound in optimizing toward the non-existing ground truth.

---

> ### Author Response · Authors · 2024-11-25
> **Post-rebuttal response to Reviewer jr8P (Part 1: Main Points)**
>
> We appreciate the reviewer's prompt reply. Their comments require a nuanced answer. We first summarize our main points:
>
> **Top-$P$ sampling and greedy decoding**:
> - The literature offers two different variants of lossy speculative decoding, one for temperature sampling ($T > 0$) and the other for greedy decoding ($T = 0$); in the paper, we have compared against the appropriate version based on $T$.
> - When $T > 0$, the proposed SpecCascade [Token] approach achieves **better trade-offs than lossy speculative decoding (non-greedy) even under Top-$P$ sampling**; in fact, the gains are **more pronounced for smaller values $P$** (new results below).
> - When $T = 0$, the proposed SpecCascade [Token] is **identical to lossy speculative decoding (greedy)**; hence, it is not meaningful to compare these methods under greedy decoding. While greedy decoding is an interesting setting, there is an entire literature on speculative sampling focused on temperature sampling (Leviathan et al., 2023; Kim et al., 2024). Indeed, one of the contributions of (Leviathan et al., 2023) over (Stern et al., 2018) was to provide a rigorous means of matching the target distribution under sampling, as opposed to greedy decoding.
>
> **Ground truth distribution**:
> - We emphasize that the “ground truth” **refers to the data generating distribution**, as commonly assumed in the literature on statistical learning and formal analysis of LLMs. **This does not mean that there is a single valid target for each input prompt**; rather, it means that each observed sequence is assumed to be a sample from some unknown probability distribution.
> - LLMs are typically pre-trained/finetuned to maximize the **log-likelihood on a sample**; so **by design** they are trained to **mimic the data-generating distribution** that the sample is drawn from.
> - Our framework is firmly grounded in the theoretical literature on cascades; the reasoning for objectives we propose are no different from those used to design prior deferral rules for cascades.
>
> We next expand on these points, and towards the end, also proactively address follow-up questions the reviewer may have. We have incorporated the contents of our response in the uploaded revised manuscript (Appendices E.2, E.7 and F).

---

> ### Author Response · Authors · 2024-11-25
> **Post-rebuttal response to Reviewer jr8P (Part 2: Experiments with Top-P Sampling)**
>
> > Additional experiments under Top-$P$ sampling
> ___
> We compare SpecCascade [Token]  and lossy speculative decoding (non-greedy version) under top-$P$ sampling with a fixed value of $P$ while varying the lenience parameter $\alpha$ to achieve different latency-quality trade-offs. We report the same evaluation metrics as in the main paper with temperature 1 and $\gamma=5$ (also included in Table 4 and Fig. 12 in the revised manuscript).
> ___
> **Reduction in latency when matching the quality of the large model:**
>
> Results on WMT:
> | Method           |  P=0.1    |  P=0.25 |   P=0.5 |
> |:-----------|:-------------|:-------------|:-------------|
> SpecDecode [Lossy, Non-greedy] | 1.55$\times$ | 1.54$\times$ | 1.64$\times$ |
> SpecCascade [Token] | **1.74$\times$** |  **1.75$\times$** | **1.73$\times$** |
>
> Results on  CNN/DM:
> | Method           |  P=0.1    |    P=0.25 |  P=0.5 |
> |:-----------|:-------------|:-------------|:-------------|
> SpecDecode [Lossy, Non-greedy] | 1.48$\times$ |  1.49$\times$ | 1.63$\times$ |
> SpecCascade [Token] | **1.61$\times$** |  1.52$\times$ | 1.65$\times$ |
> ___
>
> **Best quality metric without exceeding the latency of the large model:**
>
> Results on WMT:
> | Method           |  P=0.1    |     P=0.25 |  P=0.5 |
> |:-----------|:-------------|:-------------|:-------------|
> SpecDecode [Lossy, Non-greedy] | 27.35 |  27.34 |  27.10 |
> SpecCascade [Token] | 27.49 |  27.52 |  27.30 |
>
> Results on  CNN/DM:
> | Method           |  P=0.1    |     P=0.25 |  P=0.5 |
> |:-----------|:-------------|:-------------|:-------------|
> SpecDecode [Lossy, Non-greedy] | 15.72 |  15.71 |  15.27 |
> SpecCascade [Token] | 15.79 |  15.73 |  15.46 |
>
> **It is for the smallest value of $P$ that our proposal offers the largest gains in speed-up over lossy speculative decoding**. Unsurprisingly, both methods yield similar quality for smaller $P$ values.
>
> In fact, as shown in Figure 12, lossy speculative decoding with $P=0.1$ is able to offer only *three unique trade-off points* (despite sweeping through a fine-grained grid on $\alpha$ from $10^{-6}$ to 1); in contrast SpecCascade [Token] is able to offer a wider range of trade-off points. For example, on WMT ($P=0.1$), we have:
>
> **Reduction in latency when matching different % of large model's quality:**
> | Method           |  99% of large model    |     97% of large model |  95% of large model | 50% of large model |
> |:-----------|:-------------|:-------------|:-------------|:-------------|
> SpecDecode [Lossy, Non-greedy] | 1.55$\times$ |  1.55$\times$ |  1.55$\times$ |  2.07$\times$ |
> SpecCascade [Token] | **1.78$\times$** |  **1.88$\times$** |  **1.99$\times$** |  2.07$\times$ |
>
> ___
> ___
> **Why does lossy speculative decoding offer worse trade-offs under Top-P sampling?**
>
> From the above, the lossy speculative decoding baseline (Leviathan et al. (2022); Zhou et al., (2024)) offers worse trade-offs as $P$ gets smaller (i.e., while its quality improves with decreasing $P$, the speed-up it offers degrades).
>
> The reason for this is the acceptance criterion used in lossy speculative decoding, which under top-$P$ sampling accepts a draft token $v$ with probability:
> $$
> \min\left\\{ 1, \frac{\mathbb{T}(p)(v)}{ (1-\alpha) \cdot \mathbb{T}(q)(v)}  \right\\},
> $$
> where $\mathbb{T}(\cdot)$ truncates the distribution to only retain the top-$P$ fraction of tokens (i.e. smallest subset of tokens whose cumulative probability exceeds $P$).
>
> Notice that as $P$ gets smaller, the $\mathbb{T}(p)$ assigns zero probabilities to a majority of tokens. As a result for most draft token candidates $v$, the above criterion evaluates to 0, and the lenience parameter $\alpha$ has no effect on those tokens. Hence as $P \rightarrow 0$, the  lenience parameter $\alpha$ becomes vacuous, and thus lossy speculative decoding fails to offer meaningful trade-offs.
> ___
> ___
> **Why does our proposal offers better trade-offs under Top-P sampling?**
>
> Our proposed approach does not suffer from the same issue as it uses the lenience parameter $\alpha$ not as a scaling parameter in the acceptance criterion, but to construct a **new target distribution that is amenable to a higher acceptance rate even under top-$P$ sampling**. Specifically, in our approach, a draft token $v$ is accepted with probability:
> $$
> \min\left\\{ 1, \frac{\mathbb{T}(\pi(v))}{ \mathbb{T}(q(v))}  \right\\},
> $$
> where $\pi$ is a new target distribution defined using the lenience parameter $\alpha$ (equation 16 in paper) that interleaves between $p$ and $q$ to improve agreement to $q$ while also preserving quality.
>
> Therefore we are able to tune $\alpha$ to get a wider range of operating points and match the quality of the larger model at a lower latency.

---

> ### Author Response · Authors · 2024-11-25
> **Post-rebuttal response to Reviewer jr8P (Part 3: Special Case of Greedy Decoding)**
>
> > Response to “SpecCascade does not have a meaningful improvement in greedy decoding”
>
> ___
> As we detail below, this comparison is not meaningful for our method. What the reviewer is referring to is **Figure 5** in the paper, where we presented some results with temperature $T=0$. Notice that this figure **does not include SpecCascade [Token]**, which is the proposed method we have been discussing so far.
>
> As noted earlier, there are two variants of lossy speculative decoding in the literature:
> - what is referred to as SpecDecode [Lossy] throughout the main paper is the one proposed by Leviathan et al. (2023, A.5, page 12); Zhou et al., (2024); Tran-Thien (2023) for temperature sampling. This uses the non-deterministic acceptance criterion: $\min\left\\{ 1, \frac{p(v)}{ (1-\alpha) \cdot q(v)}  \right\\}$. This criterion fails to offer meaningful trade-offs as $T \rightarrow 0$.
> - what is referred to as lossy speculative decoding in Figure 5 is the one proposed by Leviathan et al. (see A.5, page 13, in their paper) for the special case of temperature $T=0$ (greedy decoding). Here a draft token $v$ is accepted *deterministically* when: $p(v) \geq (1 - \alpha) \cdot \max_{v’} p(v’).$
>
> We now refer to the latter as SpecDecode [Lossy, Greedy] in Figure 5. Importantly, **our proposed SpecCascade [Token] method becomes identical to SpecDecode [Lossy, Greedy] when temperature $T \rightarrow 0$** (see Lemma 9 in Appendix F). So the same plot in Figure 5 denotes both SpecDecode [Lossy, Greedy] and SpecCascade [Token].
>
> Therefore, the case of greedy decoding is not an interesting comparison to make for our method. However, as seen from the additional results we presented earlier, even with top-$P$ sampling with a small $P$ (which the reviewer deems to be a practical setting), ours is a better approach for trading-off latency vs. quality compared to lossy speculative decoding. In fact, **our approach is a better generalization of SpecDecode [Lossy, Greedy] for temperatures $T > 0$ than what is already offered in the literature**.
>
> We explain this in the newly included Appendix F and summarize the different acceptance criteria in Table 5.

---

> ### Author Response · Authors · 2024-11-25
> **Post-rebuttal response to Reviewer jr8P (Part 4: Ground-truth Distribution)**
>
> > With the recent advancement of LLMs, imitating a ground truth is not the main objective of language modeling
> ___
>
> First, we stress that the “ground-truth distribution” **does not refer to the existence of a single golden answer for every prompt**. Following the statistical learning theory literature, we use this to refer to the unknown “data generating distribution” (or “Bayes distribution”) capturing  sequences used to pre-train or fine-tune the model. For example, this could represent the distribution over plausible continuations of Wikipedia articles.
>
> Second, LLMs are typically pre-trained or finetuned to maximize the **log-likelihood on a sample**. Following convention in formal works on LLMs [1, 2, 3, 4], we assume that the pre-training and fine-tuning sequences are randomly drawn from some underlying data generating distribution; by maximizing the log-likelihood, one then attempts to make the Transformer-based language model mimic this distribution as closely as possible.  So **by design** common pre-training and fine-tuning objectives seek to **mimic the data-generating distribution** that the sample is drawn from.
>
> To reduce any confusion, we have **renamed “ground truth distribution” to “data generating distribution”** in the revised version.
>
> *References:*
>
> [1] Lotfi et al., Unlocking Tokens as Data Points for Generalization Bounds on Larger Language Models. arXiV, 2024.
>
> [2] Rajaraman et al., Toward a Theory of Tokenization in LLMs. arXiV, 2024.
>
> [3] Arora et al., A Theory for Emergence of Complex Skills in Language Models. arXiV, 2023.
>
> [4] Akinwande et al., Understanding prompt engineering may not require rethinking generalization. ICLR 2024.
> ___
> ___
> > Why sampling if the goal is to imitate the ground truth?
>
> Again, what is referred to as the ground truth is not a single golden answer for every prompt; it is simply the data generating distribution over prompt-response pairs.
>
> For a given prompt, the conditional distribution over responses need not be peaked at a single answer. So, greedy decoding need not be the default choice for inference. In applications where one desires multiple diverse responses, it is natural to sample from the modeled distribution with a temperature. In fact, in online chatbots, the user can often control the temperature and top-p parameters to tune how diverse they would like the responses to be.
> ___
>
> ___
> > Regarding the reasoning about equations 3 and 8, I do not think the theory is sound in optimizing toward the non-existing ground truth.
> ___
>
> As noted above, the “ground-truth distribution” refers to the distribution over sequences that was used to pre-train / fine-tune a LM. Such a distribution **exists by definition**!
>
> The reasoning behind Equations 3 and 8 is **similar to the rationale for minimizing a log-likelihood objective** against the data generating distribution during **pre-training / fine-tuning**. In both cases, one seeks to mimic the underlying data generating distribution.
>
> Moreover our framework is firmly rooted in the theoretical literature on cascades; **the reasoning behind the objectives we propose is no different from that for existing objectives for designing cascade deferral rules** [e.g. 5-9].
>
> *References:*
>
> [5] Gupta et al., Language Model Cascades: Token-Level Uncertainty And Beyond, ICLR 2024.
>
> [6] Jitkrittumet al., When does confidence-based cascade deferral suffice?, NeurIPS 2024.
>
> [7] Kolawole et al., Revisiting Cascaded Ensembles for Efficient Inference, arXiv, 2024.
>
> [8] Dekoninck et al., A Unified Approach to Routing and Cascading for LLMs, arXiv, 2024.
>
> [9] Mao et al., Two-stage learning to defer with multiple experts, NeurIPS 2024.

---

> > ### Author Response · Authors · 2024-11-25
> > **Post-rebuttal response to Reviewer jr8P (Part 5: Potential Follow-up Questions )**
> >
> > > Answers to potential follow-up questions the reviewer may have
> > ___
> >
> > The reviewer may have the following questions:
> > - **Can the issue with lossy speculative decoding (non-greedy) for smaller $P$ be fixed by removing the truncation operation $\mathbb{T}$ in the acceptance criterion?** Doing so will break the theoretical guarantee that (lossy) speculative decoding (Tran-Thien, 2023) offers: if our goal is to perform top-$P$ sampling on $p$, the theory requires that we apply the acceptance criterion on the same distribution we wish to sample from.
> >
> > - **Can the issue with lossy speculative decoding (non-greedy) be fixed by applying the truncation operation $\mathbb{T}$ to only the verifier $p$ and not the drafter $q$?** Unfortunately, the acceptance criterion will still suffer from the same issue as $P$ gets smaller.
> >
> > - **Should the parameter $P$ be tuned based on the lenience parameter $\alpha$?** These parameters play very different roles. In practice, $P$ is chosen based on the application at hand: we may pick a larger value when we wish to sample diverse responses for the same prompt (e.g. chat, creative writing), and a smaller value when there is a single answer we wish to elicit (e.g. QA). In contrast, the lenience parameter is chosen based on what trade-off we desire between latency and quality (a high value when there is a strict latency requirement, and a low value when quality is more important).
> >
> > - **Why not apply the deterministic acceptance criterion of SpecDecode [Lossy, Greedy]  for temperature sampling as well?**
> > This would break the guarantee we get from speculative sampling (Leviathan et al., 2023): in particular, this would have the effect of  sampling from a non-intuitive contrived target distribution. Instead, a more principled approach is the one we take in our paper, where we first design a target distribution that is suitable for the setting, and derive the appropriate acceptance criterion for it.

---

> > > ### Comment · Reviewer_jr8P · 2024-11-25
> > >
> > > I appreciate the authors' additional experiment. However, it confirms my concern. In the quality metric experiment, the performance gap between the proposed method and the baseline diminished, indicating that the quality improvement offered by the proposed method overlaps with top-p sampling. This overlap limits its effectiveness in practical settings. Additionally, since top-p sampling is applied to both the baseline and SpecCascade [Token], the SpecCascade [Token] method retains an advantage over the baseline by penalizing tokens that are not top-ranked.
> > >
> > > Regarding the authors' clarification on the data-generating distribution, I do not believe it makes a significant difference. It is likely that greedy decoding is still better than sampling for imitating the data-generating distribution.
> > > In fact, if the evaluation performed by the authors measures how well the language model mimics the data-generating distribution, greedy decoding would be a superior approach. In such a case, the proposed method is not applicable.

---

> ### Author Response · Authors · 2024-11-25
> **Re: Official Comment by Reviewer jr8P**
>
> Dear Reviewer,
>
> Thanks for the prompt reply. Your comments are based on a single operating point. However:
>
> - The central point of this paper is to trade-off latency and quality: so **a win in one of these dimensions (while being neutral on the other) is still a win**! Our approach either offers **latency reduction while being quality neutral** (low $P$), or is significantly better in both quality and latency (high $P$).
>
> - The **baseline provides no meaningful trade-offs for low $P$ values** (Fig 12); this is a **huge disadvantage** for the purposes of this paper. In fact, it becomes **mathematically vacuous** when $P$ is very small and **fails to  generate a full deferral curve.** Our proposal does not have these critical drawbacks!
>
> - Third, our method is **theoretically identical to the baseline when performing greedy decoding**.
>
> Overall, we have demonstrated that our proposal either has a significant advantage or is comparable/identical to the baseline in every setting mentioned, and also shown the baseline to have major drawbacks in some settings. We believe this demonstrates the practical value of our method.

---

### Author Response · Authors · 2024-12-02
**Common response to all reviewers**

We thank the reviewers for the elaborate comments. We have incorporated your suggestions in the manuscript (marked in blue).

Below, we summarize the main points from our **discussion with Reviewer jr8P** (Parts 1–3 of our post-rebuttal response).
___
**New experiments showing improvements with Top-$P$ sampling:**
- For **high $P$ values**, our approach is significantly **better on both quality and latency** (Table 2)
-  For **low $P$ values**, our approach offers **speed-ups in latency** while being neutral on quality (Table 4, Figure 12). Our approach being quality-neutral in this setting is not a limitation: **a win in either latency or quality (while being neutral on the other) is still a win**!
- Crucially, the **baseline provides no meaningful trade-offs for low $P$** values (Figure 12), **severely limiting its utility** for these settings. In fact, the baseline's trade-off expression is **mathematically vacuous** when $P$ is very small and fails to generate a full trade-off curve. Our proposal does not have this critical drawback! So ours is the **better method even for low $P$**.
___
**Differences to Top-$P$ sampling:**
- Top-$P$ sampling is a technique to sample a token from a fixed distribution: it is **not a mechanism to trade-off latency and quality**. In our setup, the choice of the trade-off mechanism (proposed method or the baseline) is *independent* of the choice of the sampling technique (full or top-$P$).
- Since our approach serves a **completely different purpose from top-$P$ sampling** (the two are *not* competitors), the mere use of top-ranked tokens **does not limit the utility** of our approach.
- Our goal is to **compare different trade-off mechanisms** under the **same sampling technique**. This is why when comparing to the baseline in the new results, we have both methods use top-$P$ sampling. This is not to provide an advantage to our method, but rather to ensure an **apples-to-apples comparison**!
 ___
**Applicability to greedy decoding:**
- Our proposal is **mathematically identical to the baseline under greedy decoding** (Lemma 9). So **by definition**, it is indeed applicable to greedy decoding!
- In fact, our approach is a **better generalization for temperature $T > 0$ than what the literature currently offers** as it does not suffer from the same limitation as the baseline for low temperature / $P$ values.
___
Overall, we have shown that our proposal either has a significant advantage or is comparable/identical to the baseline in every setting mentioned; we also overcome a critical drawback of the baseline in low $P$ settings. We believe this clearly demonstrates the practical value of our method.

---

### Meta-Review · Area_Chair_SnAd · 2024-12-20

**Metareview:**

This paper proposes speculative cascading, a novel integration of cascading and speculative decoding methods to improve language model inference. The approach intelligently balances speed and quality trade-offs, showing consistent improvements over existing methods under various conditions. While minor revisions are suggested for clarity and comprehensibility, the paper is sound in its contributions and warrants acceptance.

**Additional Comments On Reviewer Discussion:**

During the discussion phase, concerns were mainly raised about the fairness of baseline comparisons and the applicability to real-world scenarios, especially with respect to top-p sampling and greedy decoding. The authors provided additional experiments and clarifications, demonstrating that their method retains the advantages even under top-p sampling and is mathematically comparable to baselines in greedy decoding.

The reviewer's remaining concerns about limited improvements in certain settings reflect a call for realistic evaluation scenarios and comparisons against practical baselines. The authors have addressed these through discussion and additional experiments, thereby supporting the proposed method's efficacy.

---

### Decision · Program_Chairs · 2025-01-22

Accept (Oral)